# Modelling the spatial and temporal constrains of the GABAergic influence on neuronal excitability

**Aniello Lombardi**[ID]**, Heiko J. Luhmann, Werner Kilb**[ID]*

Institute of Physiology, University Medical Center of the Johannes Gutenberg University, Mainz, Germany

* wkilb@uni-mainz.de

**Data Availability Statement:** The source code of all models and stimulation files used in the present paper can be found in ModelDB (http://modeldb.yale.edu/267142).

## Abstract

GABA (γ-amino butyric acid) is an inhibitory neurotransmitter in the adult brain that can mediate depolarizing responses during development or after neuropathological insults. Under which conditions GABAergic membrane depolarizations are sufficient to impose excitatory effects is hard to predict, as shunting inhibition and GABAergic effects on spatio-temporal filtering of excitatory inputs must be considered. To evaluate at which reversal potential a net excitatory effect was imposed by GABA ($E_{GABA}^{Thr}$), we performed a detailed in-silico study using simple neuronal topologies and distinct spatiotemporal relations between GABAergic and glutamatergic inputs.

These simulations revealed for GABAergic synapses located at the soma an $E_{GABA}^{Thr}$ close to action potential threshold ($E_{AP}^{Thr}$), while with increasing dendritic distance $E_{GABA}^{Thr}$ shifted to positive values. The impact of GABA on AMPA-mediated inputs revealed a complex temporal and spatial dependency. $E_{GABA}^{Thr}$ depends on the temporal relation between GABA and AMPA inputs, with a striking negative shift in $E_{GABA}^{Thr}$ for AMPA inputs appearing after the GABA input. The spatial dependency between GABA and AMPA inputs revealed a complex profile, with $E_{GABA}^{Thr}$ being shifted to values negative to $E_{AP}^{Thr}$ for AMPA synapses located proximally to the GABA input, while for distally located AMPA synapses the dendritic distance had only a minor effect on $E_{GABA}^{Thr}$. For tonic GABAergic conductances $E_{GABA}^{Thr}$ was negative to $E_{AP}^{Thr}$ over a wide range of $g_{GABA}^{tonic}$ values. In summary, these results demonstrate that for several physiologically relevant situations $E_{GABA}^{Thr}$ is negative to $E_{AP}^{Thr}$, suggesting that depolarizing GABAergic responses can mediate excitatory effects even if $E_{GABA}$ did not reach $E_{AP}^{Thr}$.

## Author summary

The neurotransmitter GABA mediates an inhibitory action in the mature brain, while it was found that GABA provokes depolarizations in the immature brain or after neurological insults. It is, however, not clear to which extend these GABAergic depolarizations can contribute to an excitatory effect. In the present manuscript we approached this question with a computational model of a simplified neurons to determine what amount of a

**Funding:** This research was funded by grants of the Deutsche Forschungsgemeinschaft to WK (KI-835/3) and to HJL (CRC 1080-A01). The funders had no role in study design, data collection and analysis, decision to publish, or preparation of the manuscript.

**Competing interests:** The authors have declared that no competing interests exist.

GABAergic depolarizing effect, which we quantified by the so called GABA reversal potential ($E_{GABA}$), was required to turn GABAergic inhibition to excitation. The results of our simulations revealed that if GABA was applied alone a GABAergic excitation was induced when $E_{GABA}$ was around the action potential threshold. When GABA was applied together with additional excitatory inputs, which is the physiological situation in the brain, only for spatially and temporally correlated inputs $E_{GABA}$ was close to the action potential threshold. For situations in which the additional excitatory inputs appear after the GABA input or are distant to the GABA input, an excitatory effect of GABA could be observed already at $E_{GABA}$ substantially negative to the action potential threshold. This results indicate that even slightly depolarizing GABA responses, which may be induced during or after neurological insults, can potentially turn GABAergic inhibition into GABAergic excitation.

## 1. Introduction

The neurotransmitter γ-amino butyric acid (GABA) is the major inhibitory neurotransmitter in the adult mammalian brain [1]. GABA regulates the excitation of neurons and is thus essential for e.g. the control of sensory integration, regulation of motor functions, generation of oscillatory activity, and neuronal plasticity [2–4]. GABA mediates its effects via metabotropic $GABA_B$ receptors [5] and ionotropic $GABA_A$ receptors, ligand-gated anion-channels with a high $Cl^-$ permeability and a partial $HCO_3^-$ permeability [6]. The membrane responses caused by $GABA_A$ receptor activation thus depend on the reversal potential of GABA receptors ($E_{GABA}$), which is determined mainly by the intracellular $Cl^-$ concentration ($[Cl^-]_i$) and to a lesser extent by the $HCO_3^-$ gradient across the membrane [6].

About 30 years ago seminal studies demonstrated that $GABA_A$ receptors can mediate depolarizing and excitatory actions in the immature central nervous system [7–9]. This depolarizing GABAergic action reflects differences in $[Cl^-]_i$ homeostasis between immature and adult neurons [10–15]. In particular, low functional expression of a $K^+$-$Cl^-$ cotransporter (KCC2), which mediates the effective extrusion of $Cl^-$ and thus establishes the low $[Cl^-]_i$ required for hyperpolarizing GABAergic membrane responses [16], prevents hyperpolarizing GABA responses in the immature brain. In addition, the inwardly directed $Cl^-$ transporter NKCC1 mediates the accumulation of $Cl^-$ above passive distribution that underlies the depolarizing membrane responses upon activation of $GABA_A$ receptors [17–21]. These depolarizing GABAergic membrane responses play a role in several developmental processes [11,22,23], like neuronal proliferation [24], apoptosis [25], neuronal migration [26], dendro- and synaptogenesis [27], timing of critical periods [28] and the establishment of neuronal circuitry [29]. In addition to early development and of clinical importance, an elevated $[Cl^-]_i$ is also a typical consequence of several neurological disorders in the adult brain, like trauma, stroke or epilepsy and is considered to augment the consequences of such insults [11,30,31].

However, it is important to consider that depolarizing GABA responses do not per se lead to excitatory effects. In fact, the membrane shunting that unescapably accompanies the activation of $GABA_A$ receptors can dominate over the excitatory effects of the membrane depolarization [11,32–34]. Theoretical considerations [35,36] suggested that the relation between $E_{GABA}$ and the action potential threshold ($E_{AP}^{Thr}$) determine whether activation of $GABA_A$ receptors mediates excitatory ($E_{GABA}$ positive to $E_{Thr}^{AP}$) or inhibitory ($E_{GABA}$ negative to $E_{Thr}^{AP}$) actions. If $E_{GABA}$ was in the voltage range between resting membrane potential and $E_{Thr}^{AP}$ the activation of $GABA_A$ receptor will induce a depolarizing current, but an excitatory

postsynaptic potential (EPSP) that appears during this phase will be dampened in a way that the peak of the EPSP will reach less depolarized values. In case $E_{GABA}$ was positive to $E_{Thr}^{AP}$ the GABAergic depolarizing shift dominates over the dampening of the EPSP amplitude, leading to a more depolarized potential at the peak of the EPSP and thus an excitatory effect [36,37]. However, this concept is probably an oversimplification, as within a complex dendritic compartment the local activation of GABAergic conductance influences not only the amplitude of local EPSPs, but also the membrane length and time constants and thus temporal and spatial summation of excitatory synaptic inputs [38,39]. Moreover, the depolarizing effect of GABAergic stimulation outlasts the conductance increase associated with $GABA_A$ receptor activation, resulting in a bimodal GABA effect. Close to the initiation of GABAergic responses the shunting effect of the enhanced GABAergic conductance dominate and mediate inhibition. This phase is followed by an excitatory phase dominated by the GABAergic depolarization [40,41]. In addition, $E_{Thr}^{AP}$ is a dynamic variable, that depends on the background conductance and the density and adaptation state of voltage-gated $Na^+$ channels [10,42,43]. Experimental studies on the effects of GABAergic inputs on neuronal excitability demonstrated for immature neocortical neurons that $E_{GABA}$ required for excitatory GABAergic responses ($E_{GABA}^{Thr}$) was close to $E_{AP}^{Thr}$ [37], while in immature hippocampal neurons $E_{GABA}^{Thr}$ was considerably negative to $E_{AP}^{Thr}$ [44]. The observations that (i) the GABA effect can switch from inhibition to excitation for delayed glutamatergic inputs [40], that (ii) GABA inputs in distal dendrites can facilitate neuronal excitability [41], and that (iii) extrasynaptic GABAergic activation mediates an excitatory effect whereas synaptic inputs mediate inhibition [45], also suggest that the reversal potential required for excitation is not only defined by $E_{AP}^{Thr}$. This complexity is further supported by recent in-vivo investigations that identified excitatory as well as inhibitory effects in the immature brain [46–48]. In summary, to our knowledge no clear concept is currently available that can explain how $E_{GABA}$ influences GABAergic excitation/inhibition and the effect of GABA on spatiotemporal summation of EPSPs in the dendritic compartment.

Therefore, the present computational study investigates the dependency between $E_{GABA}$ and excitatory and inhibitory consequences of $GABA_A$ receptor activation and attempts to establish a general view of the impact of depolarizing GABAergic effects on the excitability of neurons. Our results demonstrate that only for GABAergic synapses located at or close to the soma the difference between $E_{GABA}$ and $E_{AP}^{Thr}$ predicts whether GABA has an excitatory or an inhibitory action. The $E_{GABA}$ at which depolarizing GABA actions switch from inhibition to excitation is in most cases negative to $E_{AP}^{Thr}$ and depends on the temporal and spatial relation between GABA and AMPA inputs, with a more excitatory effect on AMPA inputs that are delayed or located proximal to GABA inputs. We conclude from our results that GABA can mediate excitatory effects even if $E_{GABA}$ is considerably hyperpolarized to $E_{AP}^{Thr}$.

## 2. Results

### 2.1. Simulation of active and passive properties of immature CA3 pyramidal neurons

The parameters used for the models in this study are based on the cellular properties obtained in whole-cell patch-clamp recordings from visually identified CA3 neurons in horizontal hippocampal slices from P4-7 mice. Some parameters of these recordings have been used in our previous report [49]. The analysis of the patch-clamp experiments revealed that the immature CA3 pyramidal neurons had an average resting membrane potential (RMP) of −50.5 ± 1.3 mV, an average input resistance ($R_{Inp}$) of 1.03 ± 0.11 GOhm, and an average membrane capacity ($C_M$) of 132.3 ± 33.6 nF (all n = 42). As the passive membrane properties directly influence

synaptic integration as well as active properties, like $E_{AP}^{Thr}$ or the shape of the action potential (AP), we first adapted the spatial properties and the passive conductance $g_{pas}$ of the ball-and-stick model to emulate the recorded properties. To obtain sufficient similarity for these parameters between the model and the real cells we equipped a ball-and-stick model (soma diameter (d) = 46.6 μm, dendrite length = 1 mm, dendrite diameter = 1 μm) with a passive conductance density ($g_{pas}$) of $1.28^*10^{-5}$ nS/cm$^2$ at a reversal potential ($E_{pas}$) of −50.5 mV. This model had a RMP of -50.5 mV, a $R_{Inp}$ of 1.045 GOhm and a $C_M$ of 144.4 nF. In some experiments we reduced the topology to a simple ball model (*one node*, $d$ = 46.6 μm), without adapting $g_{pas}$, to evaluate the impact of GABA under quasi one-dimensional conditions.

With these configurations we next implemented a mechanism that provided APs with properties comparable to the APs recorded in CA3 pyramidal neuron. In particular, we were interested to simulate the AP properties around AP initiation as precisely as possible, because for the main questions of this manuscript we are interested in the $E_{AP}^{Thr}$. Since it was not possible to generate a reasonable sharp AP onset with a standard Hodgkin-Huxley (HH) model, we used a model proposed by Naundorf et al. for an optimized spike onset [50]. Using this model with an adjusted parameter set (S4 Table), we were able to simulate APs with a considerable precision (Fig 1A–1E).

Because the relation between $E_{AP}^{Thr}$ and $E_{GABA}$ is one major parameter investigated in this study and since no clear definition of the AP threshold has been given [43], we initially used 4 different methods to determine the action potential threshold (Fig 1F): 1.) The AP threshold value $E_{Thr}^{dVdt}$ was defined as the potential at which dV/dt first crosses a velocity of 10 V/s [44,51] (Fig 1F orange lines). 2.) $E_{Thr}^{d3}$ was defined as the potential at the time point of the first positive peak in $d^3V/dt^3$ [52] (Fig 1F, blue lines). 3.) $E_{Thr}^{IS}$ was determined at the intersection between linear regressions of the baseline before the AP and the rising phase of an AP (Fig 1H) [37] (Fig 1F, gray lines). 4.) $E_{Thr}^{ST}$ was defined as the maximal potential reached at the strongest subthreshold stimulation (Fig 1G, dashed line), i.e. the minimal potential that did not lead into the regenerative Hodgkin cycle. While the rheobase, i.e. the minimal suprathreshold injection current, demonstrated as expected a hyperbolic increase at shorter stimulus durations and converged to 4.2497 pA (Fig 1H), the distinct $E_{AP}^{Thr}$ parameters are virtually independent of the duration of the stimulus (Fig 1I). In the ball model average $E_{Thr}^{dVdt}$ was −35.6 mV, average $E_{Thr}^{d3}$ was −33.8 mV, average $E_{Thr}^{IS}$ was −37.9 mV, and $E_{Thr}^{ST}$ converged to −42.8 mV (Fig 1I). When using the ball-and-stick model the rheobase was slightly larger at 6.55 pA, $E_{Thr}^{dVdt}$ was −35.5 mV, $E_{Thr}^{d3}$ was −33.8 mV, $E_{Thr}^{IS}$ was −37.9 mV, and $E_{Thr}^{ST}$ converged to −42.2 mV.

For the following simulations between 55 and 63 sweeps were required for each analyzed single parameter (resulting in a total number of ca. 37000 to ca. 500000 single sweeps for each hypothesis, see Materials and Methods for details), thus a time-effective simulation was compulsory. For this purpose, we next evaluated the maximal dt interval required to obtain stable AP responses. This experiment demonstrated that the time course of AP and $E_{AP}^{Thr}$ determination remained stable up to a dt of 0.025 ms (S1 Fig). Thus we decided to use a dt of 0.025 ms in the following simulations.

## 2.2. Determination of the threshold for excitatory GABAergic responses

To identify the reversal potential at which the GABA response switches from inhibitory to excitatory, we first determined the GABAergic conductance that was sufficient to trigger an AP, which was defined as the GABAergic excitation threshold ($g_{GABA}^{Thr}$). The value of $g_{GABA}^{Thr}$ was determined by systematically increasing the conductance of a simulated GABAergic input until an AP was evoked. To determine this excitation threshold as precisely as possible,

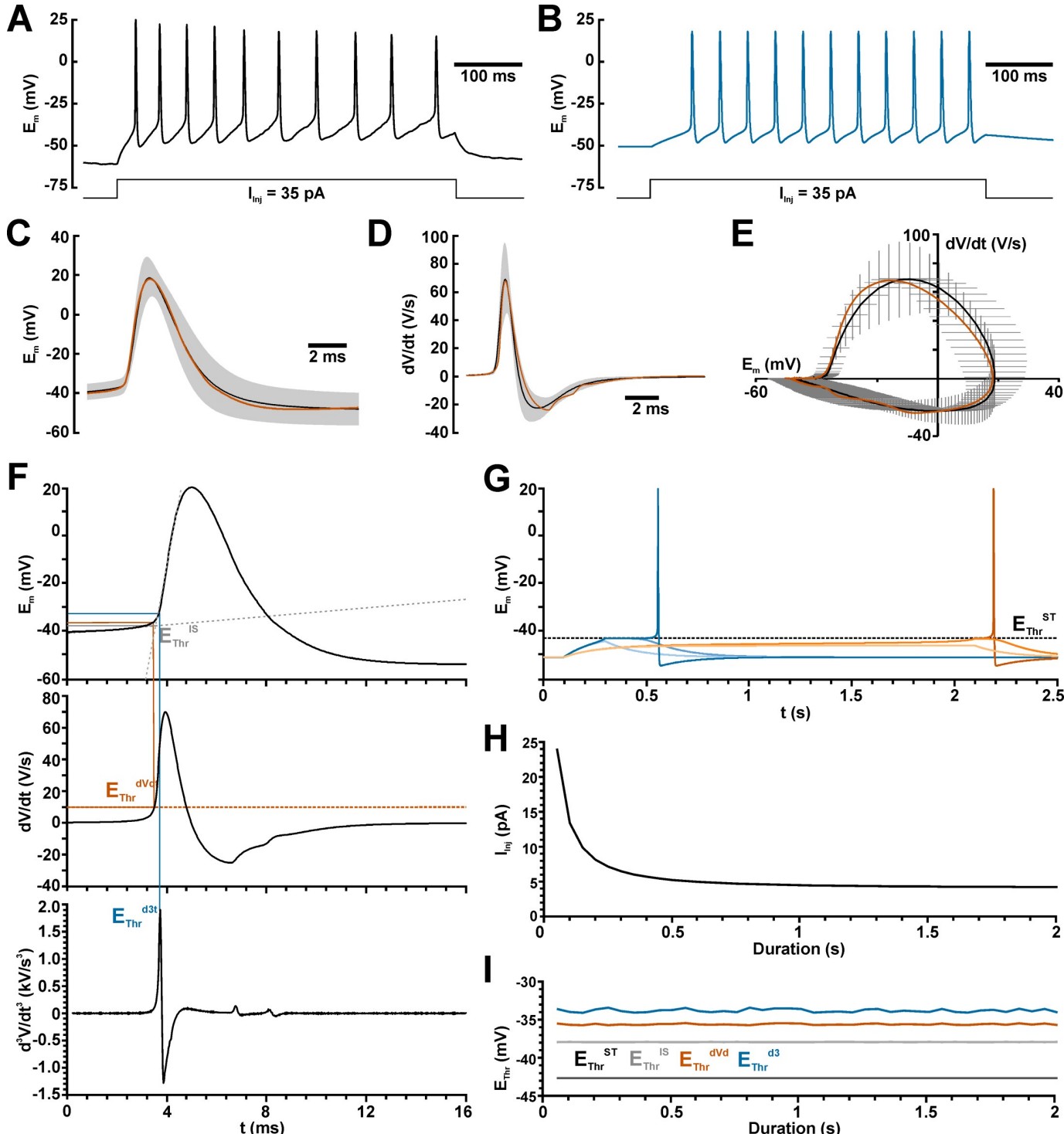

**Fig 1. Properties of recorded and simulated action potentials (APs).** A: Typical AP train recorded in a CA3 pyramidal neuron upon a current injection of +35 pA. B: AP train simulated in a ball-and-stick model using the modified Neundorf model. C: Average voltage trace of recorded APs (black line = average; gray area ± SEM) and of the simulated AP (orange trace). D: Discharge rate of recorded (black line, gray area) and simulated AP (orange trace). E: Phase plane plot of recorded (whiskers = mean ± SEM) and simulated AP (orange trace). F: Determination of the AP threshold from the intersection of linear voltage fits ($E_{Thr}^{IS}$, gray lines), from the timepoint dV/dt reaches the 10 V/s threshold ($E_{Thr}^{dVdt}$, orange lines), and from the timepoint $d^3V/dt^3$ reaches the peak value ($E_{Thr}^{d3}$, blue lines). G: Determination of the AP threshold at maximal potential of a subthreshold depolarization ($E_{Thr}^{ST}$, black lines). Blue traces indicate a 200 ms depolarizing stimulus and orange traces a 2 s stimulus. Dark tones indicate the smallest suprathreshold stimulus, middle tones the largest subthreshold stimulus and light tones a clearly subthreshold stimulus. H: Injection current ($I_{Inj}$) required to elicit an AP at different stimulus durations. I: Values of the various AP threshold parameters for different stimulation durations. Note that AP threshold is independent from the stimulation duration.

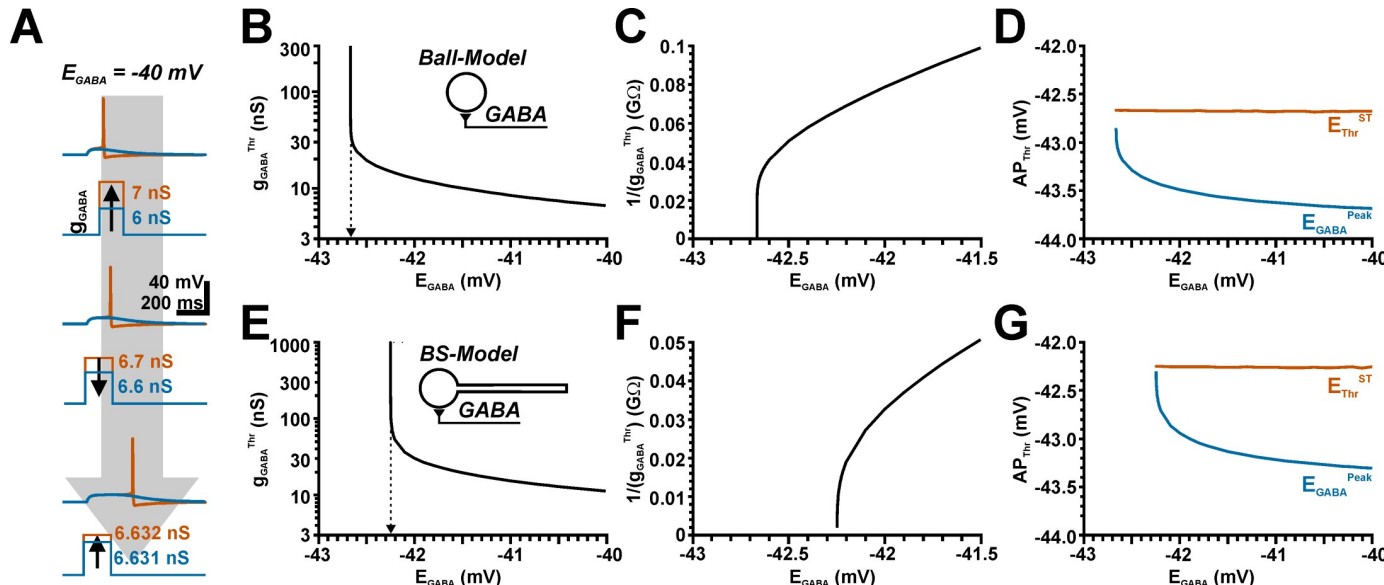

**Fig 2. Determination of the threshold conductance at different $E_{GABA}$ enable the identification of $E_{GABA}$ at which responses switch from inhibitory to excitatory ($EGABA^{Thr}$).** A: Typical voltage traces illustrating the mechanisms used to determine the threshold $g_{GABA}$ value. For this purpose, $g_{GABA}$ was increased until the first AP was induced (upper panel), then decreased by finer $g_{GABA}$ steps until the AP disappears (middle panel), followed by a subsequent increase in $g_{GABA}$ with finer $g_{GABA}$ steps (lower panel). In total, 6 alternating rounds of increased/decreased $g_{GABA}$ steps were used. The $g_{GABA}$ value required to induce an AP in the last increasing step was considered as threshold ($g_{GABA}^{Thr}$). B: Plotting $g_{GABA}^{Thr}$ versus $E_{GABA}$ demonstrate that with decreasing $E_{GABA}$ higher $g_{GABA}^{Thr}$ values were required, which approximated infinite values. C: A reciprocal plot of $g_{GABA}^{Thr}$ enables the precise determination of $E_{GABA}^{Thr}$. At $E_{GABA}$ values negative to $E_{GABA}^{Thr}$ no action potential could be induced, suggesting a stable GABAergic inhibition. D: The determined AP threshold $E_{Thr}^{ST}$ (orange line) is constant over various $E_{GABA}$, whereas the peak potential of the GABAergic depolarization, which was determined at $g_{GABA}^{Thr}$ in absence of AP mechanism ($E_{GABA}^{Peak}$, blue line) increases with decreasing $E_{GABA}$. Note that the values converged in one point when $E_{GABA}$ reaches $E_{Thr}^{ST}$. E-G: Similar plots for a ball-and-stick model. Note that $E_{GABA}^{Thr}$ was shifted to less negative values in this configuration.

we used a multi-step procedure to incrementally confine the threshold conductance (Fig 2A). This procedure was repeated for a whole set of $E_{GABA}$ values (Fig 2B).

In the ball model (*one node*, $d = 46.6$ μm) these systematic simulations demonstrated an obvious hyperbolic increase of $g_{GABA}^{Thr}$ when $E_{GABA}$ approaches values below −42.5 mV (Fig 2B). The $g_{GABA}^{Thr}$ curve approximated an $E_{GABA}$ of −42.67 mV, which was precisely determined from a reciprocal plot of the $g_{GABA}^{Thr}$ values (Fig 2C). Negative to an $E_{GABA}$ of −42.67 mV no action potential could be evoked, regardless of the amount of $g_{GABA}$. These $E_{GABA}$ values thus reflects the threshold, at which GABA actions can mediate a direct excitation and we termed this value "threshold $E_{GABA}$" ($E_{GABA}^{Thr}$). Note that this value is in the range of the $E_{Thr}^{ST}$ value of −42.8 mV determined in the previous experiments. Since $E_{AP}^{Thr}$ is influenced directly by the total membrane conductance, we also determined the amplitude of the GABAergic voltage response under conditions when the AP initiation was blocked ($E_{GABA}^{Peak}$) as well as the different $E_{AP}^{Thr}$ parameters. These analyses revealed that $E_{Thr}^{d3}$ was around −33.5 mV for all $E_{GABA}$. $E_{Thr}^{ST}$ was relatively stable around −42.7 mV (Fig 2D). $E_{GABA}^{Peak}$ was for higher $E_{GABA}$ around −43.7 mV and showed a positive shift with decreasing $E_{GABA}$ that converged to values of −42.8 mV (Fig 2D).

In summary, these results indicate that GABA acts as excitatory neurotransmitter as long as $E_{GABA}$ is positive to −42.67 mV, which is extremely close to the AP threshold $E_{Thr}^{ST}$. This observation is in line with previous predictions that propose exactly this relation between $E_{AP}^{Thr}$ and $E_{GABA}$ [35,36]. In addition, our simulations suggest that $E_{Thr}^{ST}$ is probably the most relevant definition for $E_{AP}^{Thr}$ if the direction of GABA effects should be predicted from the difference between $E_{GABA}$ and $E_{AP}^{Thr}$.

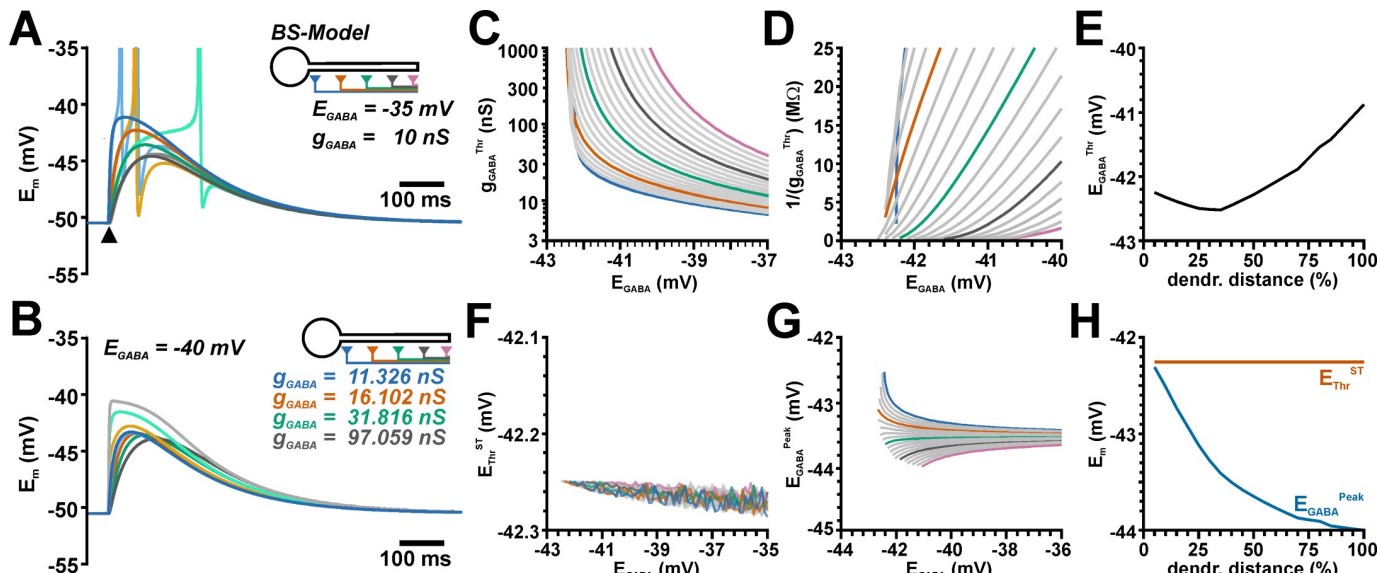

**Fig 3. Determination of EGABA$^{Thr}$ at different dendrite positions.** A: Simulated voltage traces obtained with the given parameters at different locations as indicated by color code. The light colored traces represent simulation with added AP mechanism. The amplitude of GABA responses clearly depends on the dendritic location. B: Simulated voltage traces for $g_{GABA}^{Thr}$ and $E_{GABA}$ of −40 mV at the soma (dark colors) and the synaptic site (light colors). For each location different $g_{GABA}$ (as indicated) had to be used. Note that at distant synapses considerable large $g_{GABA}$ were required, which virtually clamped $E_m$ at the synaptic site to $E_{GABA}$. C: Systematic plot of $g_{GABA}^{Thr}$ determined at various $E_{GABA}$. The curves were obtained from 20 equidistant positions along the dendrite. The 1$^{th}$, 5$^{th}$, 10$^{th}$, 15$^{th}$ and 20$^{th}$ trace is color-coded as in A for better readability. D: The reciprocal plot of $g_{GABA}^{Thr}$ revealed that the curves did not monotonically approach the abscissa. Therefore, $E_{GABA}^{Thr}$ was estimated from a linear fit to the last two data-points. E: $E_{GABA}^{Thr}$ showed a considerable shift towards depolarized potentials with increasing dendritic distance. F: The AP threshold $E_{Thr}^{ST}$ remained rather stable with different $E_{GABA}$ or different synaptic location. G: The peak potential ($E_{GABA}^{Peak}$) of the somatic GABAergic depolarization at $g_{GABA}^{Thr}$ converges toward $E_{Thr}^{ST}$ only for soma-near synapses (dark blue trace). With more distant synapses less depolarized $E_{GABA}^{Peak}$ was required. Color code as in C. H: While the average $E_{Thr}^{ST}$ (orange line) is stable for all dendritic locations, the $E_{GABA}^{Peak}$ at threshold stimulation (blue line) is shifted to more negative values with increasing dendritic distance.

Next we performed the same simulation with a ball-and-stick model. These simulations revealed that the $g_{GABA}^{Thr}$ curve approximated an $E_{GABA}$ of −42.2 mV (Fig 2E and 2F), which is very close to the $E_{Thr}^{ST}$ (−42.2 mV) determined for the ball-and-stick model. $E_{Thr}^{d3}$ was around −33.5 mV for all $E_{GABA}$. $E_{Thr}^{ST}$ was stable at values around −42.2 mV and converges at low $E_{GABA}$ to −42.25 mV (Fig 2G). $E_{GABA}^{Peak}$ was for higher $E_{GABA}$ around −43.4 mV and converged with decreasing $E_{GABA}$ to −42.3 mV (Fig 2G). Thus, $E_{GABA}^{Thr}$ for a somatic synapse is still in good agreement with the AP threshold value $E_{Thr}^{ST}$ with this slightly more complex neuronal topology.

For the next set of experiments, we located a single GABA synapse along the dendrite of the ball-and-stick model and determined $E_{GABA}^{Thr}$ for each of these 20 synaptic positions, using the method described above. The considerable conductance and capacitance provided by the dendritic membrane leads, as expected, to a reduced amplitude and a slower time course of the GABAergic PSPs recorded at the dendritic positions (Fig 3A). Accordingly, larger $g_{GABA}$ values were required to trigger APs for more distant dendritic locations of GABAergic inputs (Fig 3B and 3C). At the most distant dendritic positions $g_{GABA}$ values above 100 nS (i.e. more than 100x of $g_{GABA}$ of a single synaptic event [49]) were required to trigger an AP, which virtually clamped the dendritic membrane at the synapse position to $E_{GABA}$ (Fig 3B). A systematic analysis of $g_{GABA}^{Thr}$ at different $E_{GABA}$ values illustrated that $g_{GABA}^{Thr}$ showed a considerably less steep dependency on $E_{GABA}$ at more distant dendrite positions (Fig 3C). The reciprocal plot of $g_{GABA}^{Thr}$ demonstrated that the $g_{GABA}^{Thr}$ values did not converge at similar $E_{GABA}$ values for the different synapse locations, but that the curves reached the abscissa at considerable more

positive values for distant GABAergic inputs (Fig 3D). Intriguingly, the synapses close to the soma revealed a $E_{GABA}^{Thr}$ value close to $E_{Thr}^{ST}$, which was shifted to slightly more negative $E_{GABA}^{Thr}$ values for dendritic synapses at a distance of ca. 250 μm, and then increased to positive values with additional distance to the soma (Fig 3E). $E_{GABA}^{Peak}$, which was determined in the absence of AP mechanisms and reflects the effective voltage fluctuation at the soma and thus the AP initiation site, was shifted to negative potentials at more distant dendritic positions (Fig 3G and 3H), while the position of GABA synapses had no major effect on $E_{Thr}^{ST}$ (Fig 3F and 3H). In summary, these simulations revealed that $E_{GABA}^{Th}$ is not close to the AP threshold value $E_{Thr}^{ST}$ for synapses that are located in the dendrite, but that $E_{GABA}^{Th}$ is shifted to more positive values with increasing distance. This observation suggests that for dendritic synapses a more positive $E_{GABA}$ (corresponding to a higher $[Cl^-]_i$) is required to mediate a direct excitatory effect.

## 2.3. Effect of phasic GABAergic inputs on glutamatergic excitation

The previous results demonstrated that only at perisomatic synapses $E_{GABA}^{Thr}$ was reached when $E_{GABA}$ was at the action potential threshold $E_{Thr}^{ST}$, but that $E_{GABA}^{Thr}$ was systematically shifted to positive $E_{GABA}$ at distant synapses in a ball-and-stick model. However, these experiments do not reflect the physiological situation of GABAergic transmission in the brain. First, the threshold conductance $g_{GABA}^{Thr}$ determined by these simulations is clearly above physiological values for moderate GABAergic inputs [49,53,54], making a direct excitatory GABAergic input implausible. And second, synaptic activity is characterized by the co-activation of GABA and glutamate receptors [55–57], with the latter constituting the main excitatory drive [58]. Therefore, we next simulated the impact of a GABAergic co-stimulation on glutamatergic synaptic inputs and determined the $g_{AMPA}$ values that were required to trigger an AP. For the present simulation we used a simplified model of glutamatergic synaptic inputs, neglecting NMDA receptors [59,60]. We considered to use the reduced model containing only AMPA and GABA receptors to ease the interpretation of the interactions between both synaptic inputs. The functional relevance of both, $GABA_A$ [8,61–63] and AMPA [64–66] receptors from early postnatal stages into adulthood has been clearly demonstrated in the hippocampus and neocortex.

As in the previous experiments, we varied $E_{GABA}$ to determine $E_{GABA}^{Thr}$, which is defined as the $E_{GABA}$ value at which the GABAergic effect shifts from inhibitory (i.e. GABA co-activation requires larger $g_{AMPA}$ to trigger APs) to excitatory action (i.e. GABA co-activation requires less $g_{AMPA}$) (Fig 4A). This effect was quantified as the GABAergic excitability shift ($\Delta g_{AMPA}^{Thr}$), with $g_{AMPA}^{Thr}$ describing the $g_{AMPA}$ value sufficient to trigger an AP, and $\Delta g_{AMPA}^{Thr}$ defined as difference in $g_{AMPA}^{Thr}$ between conditions with and without GABAergic co-stimulation: $[\Delta g_{AMPA}^{Thr} = (g_{AMPA}^{Th})_{withGABA} - (g_{AMPA}^{Th})_{w/oGABA}]$.

In the first set of experiments we simulated the effect of GABA pulses provided synchronously with AMPA inputs in a ball model (Fig 4A) using a constant $g_{GABA}$ of 3.95 nS. These experiments demonstrated that the co-stimulation of a GABAergic input can attenuate or enhance AP triggering upon glutamatergic stimulation, depending on $E_{GABA}$ (Fig 4A). As expected, such a GABA co-stimulation enhanced $g_{AMPA}^{Thr}$ at hyperpolarized $E_{GABA}$, while smaller $g_{AMPA}^{Thr}$ values were required at more depolarized $E_{GABA}$ (Fig 4B). From the intersection of this $g_{AMPA}^{Thr}$ with the $g_{AMPA}^{Thr}$ recorded in the absence of GABAergic inputs we determined that $E_{GABA}^{Thr}$ amounted to −44.4 mV under this condition (Fig 4B), which is considerable more negative than $E_{Thr}^{ST}$ of −42.8 mV determined in the ball model. Additional experiments with different $g_{GABA}$ values revealed that $E_{GABA}^{Thr}$ depends on $g_{GABA}$ (Fig 4C– 4E). However, only at rather large $g_{GABA}$ values $E_{GABA}^{Thr}$ approached toward values > −44

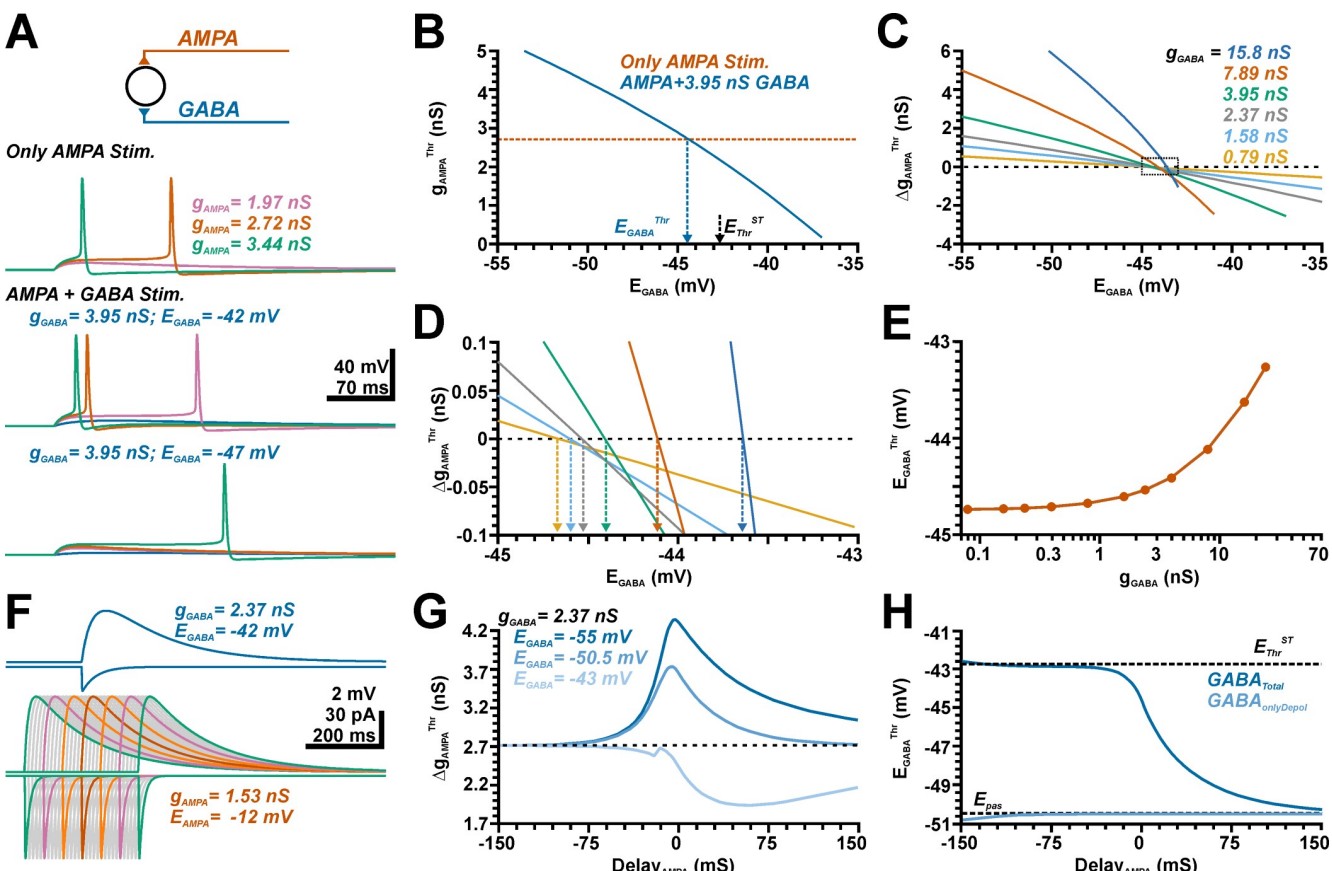

**Fig 4. Influence of a GABAergic input at different EGABA$^{Thr}$ on the AMPA receptor-dependent excitation threshold.** A: Simulated voltage traces illustrating the membrane responses induced by three different conductances of the AMPA synapse in the absence (top traces) and the presence of a simultaneous GABAergic input at $E_{GABA}$ of −42 mV (middle traces) and −47 mV (lower traces). B: Plot of the minimal $g_{AMPA}$ required to trigger an AP ($g_{AMPA}^{Thr}$) versus the $E_{GABA}$ of the synchronous GABA input ($g_{GABA}$ = 3.95 nS). The $E_{GABA}$ value at which this curve intersects with $g_{AMPA}^{Thr}$ determined in the absence of GABA (orange line) defines the GABA concentration at which GABA switches from excitatory to inhibitory ($E_{GABA}^{Thr}$). C: Plot of $\Delta g_{AMPA}^{Thr}$ versus $E_{GABA}$ for different $g_{GABA}$ values, as indicated in the graph. D: A magnification of the marked area in C allows the determination of $E_{GABA}^{Thr}$ for the different $g_{GABA}$, color code as indicated in C. E: Plot of the $E_{GABA}^{Thr}$ determined at different $g_{GABA}$. Note that $E_{GABA}^{Thr}$ is substantially negative to $E_{Thr}^{ST}$ and increases at higher $g_{GABA}$. F: Simulation of membrane currents (downward deflections) and membrane changes (upward deflections) upon a GABAergic (blue traces) and glutamatergic stimulation. The lower traces represent glutamatergic inputs shifted by ± 150 ms in 10 ms steps, each 5th trace was colored for better readability. Note that the depolarization shift outlasts the conductance shift for both inputs. G: Influence of the timing between AMPA and GABA input on $\Delta g_{AMPA}^{Thr}$ determined at 3 exemplary $E_{GABA}$. Note that the maximal inhibitory effect at hyperpolarizing ($E_{GABA}$ = −55 mV) or pure shunting GABAergic inputs ($E_{GABA}$ = −50.5 mV) were observed for synchronous AMPA inputs, while the excitatory influence of GABA at depolarized $E_{GABA}$ of −43 mV was maximal for substantially delayed AMPA inputs. H: Quantification of $E_{GABA}^{Thr}$ (dark blue) for different delays between GABA and AMPA inputs. Note that for AMPA inputs preceding GABA inputs $E_{GABA}^{Thr}$ was close to the AP threshold, while for AMPA inputs lagging GABA inputs $E_{GABA}^{Thr}$ approximated −50.5 mV. The light blue traces represent $E_{GABA}^{Thr}$ determined for pure simulated GABAergic depolarizations which persistently results in a $E_{GABA}^{Thr}$ close to −50.5 mV.

mV. At lower, physiologically more relevant $g_{GABA}$ values $E_{GABA}^{Thr}$ converges to a value of −44.7 mV (Fig 4E). This observation indicates that $E_{GABA}^{Thr}$ was consistently lower than $E_{Thr}^{ST}$, implying that GABAergic inputs are under these conditions more excitatory than expected from the difference between $E_{GABA}$ and $E_{AP}^{Thr}$.

Is has already been proposed that the GABAergic depolarization outlasts the GABAergic currents and can add an additional excitatory drive to neurons [40]. Our simulations replicated this typical behavior, both GABAergic and glutamatergic membrane depolarization outlasted the time course of the respective currents (Fig 4F), To investigate whether the systematic shift of $E_{GABA}^{Thr}$ towards more hyperpolarized potentials was indeed caused by the differential

impact of GABAergic conductance and GABAergic membrane depolarization on the AMPA-mediated excitation, we systematically advanced or delayed the time point of AMPA inputs (Fig 4F). These simulations revealed that, as expected, the strongest inhibitory effect of a GABAergic input for both hyperpolarizing (at $E_{GABA} <$ RMP) and shunting inhibition (at $E_{GABA} =$ RMP) was observed when it was synchronous to the glutamatergic input (Fig 4G). In contrast, at more depolarized $E_{GABA}$ the maximal excitatory effect occurred when the AMPA input was given about 60 ms after the GABA input (Fig 4G, light trace), i.e. at a time point when the GABAergic conductance virtually ceased but a considerable GABAergic depolarization persisted (Fig 4F, blue traces). A systematic determination of $E_{GABA}^{Thr}$ for different delays demonstrated that $E_{GABA}^{Thr}$ was relatively stable around −43 mV for APMA inputs that preceded GABA inputs, and was thus close to $E_{Thr}^{ST}$ (Fig 4H). In contrast, with increasing delays of the glutamatergic inputs $E_{GABA}^{Thr}$ converged to −50.5 mV, i.e. to the RMP determined by the reversal potential of the passive membrane conductance (Fig 4H). In summary, these findings suggest (i) that at preceding AMPA inputs the influence of GABA on this glutamatergic input was dominated by the GABAergic conductance change and thus converged to $E_{Thr}^{ST}$ and (ii) that at delayed glutamatergic inputs the influence of GABA on this glutamatergic input was dominated by the GABAergic depolarization.

In the absence of a GABAergic conductance shift each depolarization above −50.5 mV should reduce the distance to the $E_{AP}^{Thr}$ and should thus impose an excitatory effect. To verify this hypothesis, we recorded the GABAergic currents at different $E_{GABA}$ and replayed these currents to the modelled neurons via I-clamp, thereby isolating the effect of the GABAergic depolarization from the GABAergic conductance shift. Indeed, these simulations demonstrated that the effect of the pure GABAergic depolarization reversed at an $E_{GABA}$ of −50.5 mV (Fig 4H, light trace).

In summary these experiments demonstrated that the effect of a GABAergic stimulus on glutamatergic synaptic inputs cannot simply be predicted from the difference between $E_{GABA}$ and the $E_{AP}^{Thr}$ threshold, but that, depending on the temporal relation between GABAergic and glutamatergic inputs, $E_{GABA}$ is substantially lower than $E_{AP}^{Thr}$ and thus GABA acts more excitatory than expected from the $E_{GABA}$ to $E_{AP}^{Thr}$ relation.

In the next set of experiments, we evaluated how the spatial relation between GABAergic and glutamatergic inputs affects $E_{GABA}^{Thr}$ in a ball-and-stick model. For these simulations, we systematically varied both, GABA and AMPA synapse along the dendrite, using 20 equidistant positions each (Fig 5A), and stimulated both synapses synchronously. Simulations of single inputs revealed that the time course of the glutamatergic and GABAergic depolarizations critically depended on the dendritic location (Fig 5A), which reflect spatial filtering [67]. To prevent that this temporal scattering affects the spatial analysis of GABA/AMPA relations, we determined the maximum of the depolarization in control sweeps performed before each run of the definite simulation for each combination of $g_{AMPA}$, AMPA location, $E_{GABA}$, and GABA location in the absence of an AP mechanism. For the definite simulation sweep the temporal relation between glutamatergic and GABAergic input was shifted such that peak depolarization of GABA and AMPA responses coincided (Fig 5A). To get an impression how a depolarizing GABAergic input at different locations influences $g_{AMPA}^{Thr}$, we first varied the position of a GABAergic synapse with a $g_{GABA}$ of 7.89 nS and an $E_{GABA}$ of −40 mV along the dendrite and determined $g_{AMPA}^{Thr}$ for each of the 20 AMPA synapse along the dendrite (Fig 5B). These simulations showed, as expected, that (i) $g_{AMPA}^{Thr}$ increased with increasing dendritic distance, and (ii) that for a soma-near GABAergic synapse the excitatory effect of GABA was stronger than for distal dendritic locations, as indicated by the larger $g_{AMPA}^{Thr}$ required for the distal GABA synapses (Fig 5B). However, we could also demonstrate that (iii) the slope of the $g_{AMPA}^{Thr}$ became shallower for AMPA inputs distal to the GABA inputs (Fig 5B), indicating a

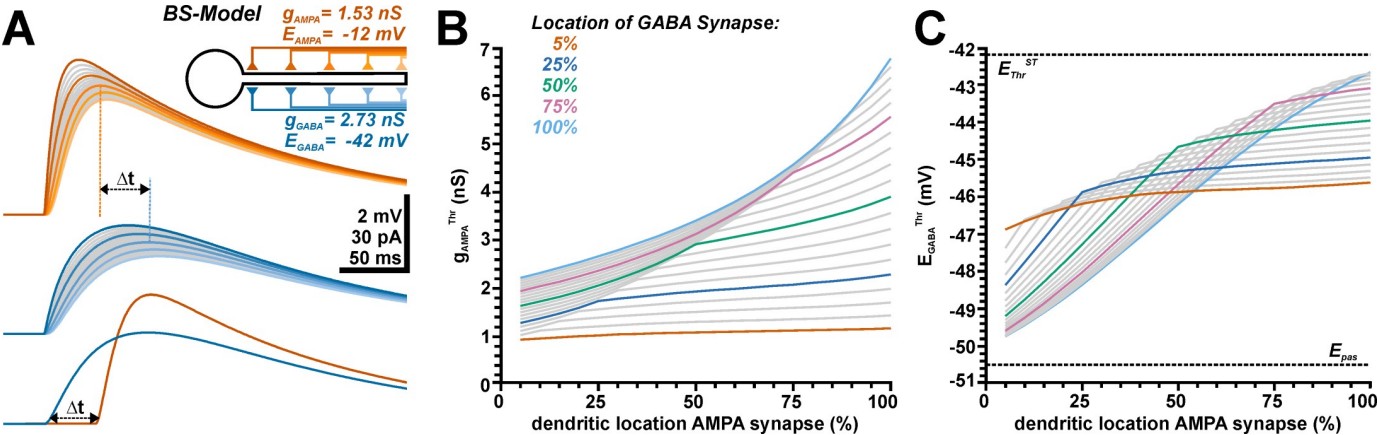

**Fig 5. Influence of the spatial relation between the AMPA receptor-dependent and the GABA receptor-dependent synaptic input on gAMPA$^{Thr}$ and EGABA$^{Thr}$.** A: Simulated voltage traces illustrating the membrane responses induced by AMPA synapses (orange traces) and by GABA synapses (blue traces) located at different dendritic locations. The colored traces represent synapses at 5%, 25%, 50%, 75% and 100% of the dendritic length, as color-coded in the schematic inset. Note the slower onset kinetics and delayed peak for distant dendritic synapses. The lower traces depict how the delay of GABA and AMPA was adjusted to obtain synchronous peak depolarizations. B: Effect of the dendritic location on $g_{AMPA}^{Thr}$ simulated for 20 equidistant positions of the GABAergic synapse ($g_{GABA}$ = 7.89 nS; $E_{GABA}$ = −40 mV). Each line represents the results for one GABA synapse position, the color code identifies every 5th position as indicated. Note the shallow dependency of $\Delta g_{AMPA}^{Thr}$ for proximal and the steep dependency for distal GABA synapses. C: Dependency of $E_{GABA}^{Thr}$ on the dendritic positions of AMPA synapses, each line represents the results for one GABA synapse position, with shade coding as in B. Note the shallow location dependency with $E_{GABA}^{Thr}$ between ca. −46 and −47 mV for the proximal GABA synapses, while for distal GABA synapses a steep $E_{GABA}^{Thr}$ profile between ca. -43 mV and -50 mV was observed.

strong non-linear influence of GABAergic inputs. To determine how the spatial relation between glutamatergic and GABAergic inputs affects $E_{GABA}^{Thr}$ we subsequently varied $E_{GABA}$ (at $g_{GABA}$ of 7.89 nS) for all combinations of synaptic positions and determined when $\Delta g_{AMPA}^{Thr}$ switches the direction (Fig 5C). These simulations revealed a complex relation between these three parameters. If the GABAergic synapse was located in the proximal dendrite close to the soma $E_{GABA}^{Thr}$ was only weakly dependent on the site of the AMPA synapse and amounted to values between ca. −46 mV and −47 mV (Fig 5C, orange trace). If the GABA synapse was located more distally $E_{GABA}^{Thr}$ showed a step dependency on the location of the AMPA synapse for all AMPA synapses located proximally to the GABA synapse, while the shallow dependency was maintained for the more distal synapses (Fig 5C). Under this condition $E_{GABA}^{Thr}$ approached −50 mV for proximal AMPA synapses, i.e. when both synapses were 950 μm apart and thus the GABAergic depolarization dominated over the more local shunting effect (see light blue trace in Fig 5C). In contrast, for distal AMPA and GABA synapses, which represent spatially correlated inputs distant from the AP initiation zone, $E_{GABA}^{Thr}$ approached $E_{Thr}^{ST}$ (Fig 5C).

GABAergic synapses are not only located in the somatodendritic compartment, but can also be found in the axon initial segment [68]. Intriguingly at these synapses the developmental profile of Cl$^-$ transporter expression was extended until peri-adolescent periods [69], resulting in a substantially depolarized $E_{GABA}$ of this GABA synapse with putative excitatory effect [70,71]. To estimate whether the strategic location of this synapse implies a specific dependency on $E_{GABA}$, we also simulated a simple topology that includes an axon with a GABAergic synapse in its initial segment (S2 Fig). For somatically located AMPA inputs these simulations revealed that $E_{GABA}^{Thr}$ amounted to −44.1 mV (S2A and S2B Fig) for a physiological $g_{GABA}$ of 0.789 nS. With increasing $g_{GABA}$ $E_{GABA}^{Thr}$ was obtained at more depolarized values (S2B Fig). For dendritic localizations of AMPA synapses we found that $E_{GABA}^{Thr}$ was shifted to more negative values with increasing distance of the synapses from the soma (S2C Fig). However,

these $E_{GABA}^{Thr}$ values were under all conditions less than 1 mV depolarized to simulations with identical parameters but with a somatic localization of the GABA synapse. Thus with regard to their $E_{GABA}$ dependency GABA synapses in the axon initial segment are comparable to somatic GABA synapses.

In summary, these results demonstrate that both, the spatial relation between GABAergic and glutamatergic synapses as well as the location of the GABA synapse influences $E_{GABA}^{Thr}$. However, only for spatially correlated inputs at distal dendrites $E_{GABA}^{Thr}$ was close to the $E_{AP}^{Thr}$. With increasing distance between both synapses and with a closer approximation of the GABA synapse to the soma, $E_{GABA}^{Thr}$ was shifted to more negative values, again indicating that GABA mediates a more prominent excitatory action than expected from the difference between $E_{GABA}$ and $E_{AP}^{Thr}$.

## 2.4. Effect of tonic GABAergic inputs on glutamatergic excitation

GABA influences neuronal excitability not only via synaptic inputs, but also extrasynaptic, tonic GABAergic currents substantially contribute to the GABAergic effects [72,73] and can mediate even excitation during development [45]. Therefore, we next analyzed how a tonic GABAergic conductance ($g_{GABA}^{tonic}$) influences $g_{AMPA}^{Thr}$ and $E_{GABA}^{Thr}$ in a ball model (Fig 6A), using a $g_{GABA}^{tonic}$ between 87.5 pS/cm$^2$ and 8.75 μS/cm$^2$, corresponding to values from 1/100 to 1000 times of the experimentally determined tonic GABA conductance of 8.75 nS/cm$^2$ [53]. These experiments demonstrated, that $g_{GABA}^{tonic}$ can attenuate or enhance AP induction by AMPA synapses, depending on $E_{GABA}$. As expected, the slope of the GABAergic influence increased with $g_{GABA}^{tonic}$ (Fig 6A). And as expected, tonic GABAergic conductance enhanced $g_{AMPA}^{Thr}$ at hyperpolarized $E_{GABA}$, while smaller $g_{AMPA}^{Thr}$ values were required at more depolarizied $E_{GABA}$ (Fig 6A). From the intersection of these $g_{AMPA}^{Thr}$ with the basal $g_{AMPA}^{Thr}$ (obtained in the absence of tonic GABA), $E_{GABA}^{Thr}$ was determined (Fig 6B). Notably, these $E_{GABA}^{Thr}$ were rather constant at ca. −47.5 mV within a wide range of $g_{GABA}^{tonic}$, spanning from 0.01 to about the experimentally determined $g_{GABA}^{tonic}$ value. Only at very high $g_{GABA}^{tonic}$ of > 100 nS/cm$^2$ $E_{GABA}^{Thr}$ approached $E_{Thr}^{ST}$ (which is under this conditions shifted to positive values due to the massively enhanced total membrane conductance). In summary, these results indicate that tonic GABAergic conductances can mediate an excitatory effect even if $E_{GABA}$ was substantially negative to $E_{AP}^{Thr}$.

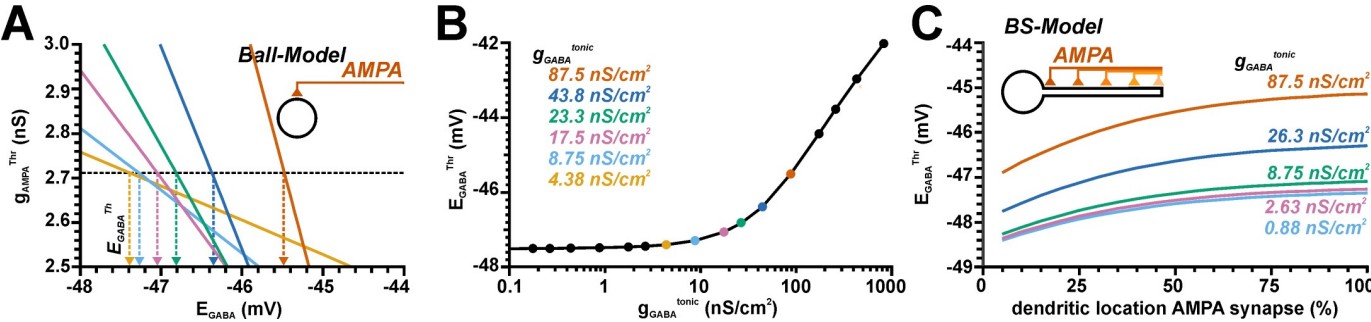

**Fig 6. Influence of tonic GABAergic conductances on the AMPA receptor-dependent excitability in a simple ball and a ball-and-stick model.** A: Plot of $g_{GABA}^{Thr}$ at different $E_{GABA}$. The colored lines represent different tonic $g_{GABA}$ values as indicated in B. The increased slope of the curves with higher $g_{GABA}^{tonic}$ illustrates the higher inhibitory effect under this conditions. From the intersection of the plots with the $g_{AMPA}^{Thr}$ value obtained in the absence of tonic GABA (dashed line) the $E_{GABA}^{Thr}$ values were determined. B: Plot of $E_{GABA}^{Thr}$ determined at different $g_{GABA}^{tonic}$. Note that $E_{GABA}^{Thr}$ is negative to $E_{Thr}^{ST}$ for $g_{GABA}^{tonic}$ < ca. 300 nS/cm$^2$. C: Influence of different dendritic locations of AMPA synapses in a ball-and-stick model on the AMPA receptor-dependent excitability determined for different $g_{GABA}^{tonic}$. Note the substantial shift of $E_{GABA}^{Thr}$ to positive values with more distant AMPA synapses and the systematic depolarized shift with increasing $g_{GABA}^{tonic}$.

In addition, we investigated how the $E_{GABA}$ of $g_{GABA}^{tonic}$ affects the excitation generated by AMPA synapses located along the dendrite in a ball-and-stick model (Fig 6C). These simulations revealed that $E_{GABA}^{Thr}$ was systematically shifted to positive values for distal AMPA synapses and that $E_{GABA}^{Thr}$ was more positive for larger $g_{GABA}^{tonic}$ at all dendritic positions (Fig 6C). These observations suggest that a tonic GABAergic conductance mediates an excitatory effect even at $E_{GABA}$ that is substantially negative to $E_{AP}^{Thr}$, but that an inhibitory effect of tonic GABAergic conductance is higher at distal AMPA-mediated inputs.

## 2.5. Effects of GABAergic inputs on glutamatergic excitation in neurons with a realistic dendritic morphology

So far our results demonstrated that the impact of $E_{GABA}$ critically depends on the location of GABAergic synapses and their distance to glutamatergic inputs. However, since our previous models represent rather simplified morphological conditions, we next attempted to estimate the impact of distinct $E_{GABA}$ values under more realistic conditions. For this purpose, we used a neuronal topology we derived from a reconstructed, biocytin-labeled CA3 pyramidal neuron (Fig 7A and 7B) [49], that we already utilized in previous in-silico studies [74,75]. Using this topology, we estimated the effect of GABA synapses at different $E_{GABA}$ from the impact of GABAergic input on the spike probability upon glutamatergic inputs. GABA and AMPA synapses were randomly distributed across the dendritic compartment and each synapse was stimulated at a random time point during the 2 s stimulation interval (Fig 7B and 7C). The spike probability ($p_{AP}$) was determined from 999 single sweeps, each with a new distribution of synapse location and stimulus times.

We first determined the $g_{AMPA}$ values required to mediate a $p_{AP}$ of 50% ($g_{AMPA}^{50}$) in the absence of GABAergic synapses (Fig 7). From these simulations, we obtained $g_{AMPA}^{50}$ values between 0.92 nS and 0.275 nS for stimulation frequencies between 1 Hz and 20 Hz (Fig 7F, S1 Table). In order to reveal the spatial components of GABAergic inhibition in further experiments, we repeated this $g_{AMPA}^{50}$ determination also for AMPA receptors that were distributed only in distal or proximal dendrites (Fig 7G and 7H, S1 Table).

Next we randomly distributed and stimulated GABA and AMPA synapses across the dendritic compartment (Fig 8A), using the previously determined $g_{AMPA}^{50}$ values as gain of the AMPA inputs. As expected, these simulations revealed that random co-stimulation with GABAergic synapses reduced $p_{AP}$ at negative $E_{GABA}$, while $p_{AP}$ was enhanced at more positive $E_{GABA}$ (Fig 8B and 8C). The $E_{GABA}^{Thr}$ values obtained from the intersection of the $p_{AP}$ curve with the $p_{AP}$ value of AMPA inputs only (which was close to 0.5, but not exactly at this value, see the dashed lines in Fig 8D), was for a frequency of 20 Hz at −43.5 mV (Fig 8D, S2 Table), and thus only ca 1 mV positive to the $E_{Thr}^{ST}$ value of −42.8 mV. At lower frequencies, $E_{GABA}^{Thr}$ was substantially more negative and reached −45.9 mV at 1 Hz (Fig 8D, S2 Table), indicating that under these conditions an excitatory effect of GABA can be observed already at lower $[Cl^-]_i$.

Since GABAergic synapses showed a non-homogenous distribution in neurons [68], we also simulated conditions in which the GABA synapses were randomly distributed either in the most distal or the most proximal dendrites (Fig 9). In these simulations we placed the AMPA synapses either throughout the whole dendrite (Fig 9A and 9E), or opposing to the GABA synapse location (Fig 9C and 9G). These simulations revealed that with distally located GABA synapses $E_{GABA}^{Thr}$ was slightly shifted to negative values by ca. 0.1–0.5 mV (Fig 9A and 9B; S2 Table). This effect was not systematically altered when the AMPA synapses were restricted to the proximal dendrite (Fig 9C and 9D; S2 Table). Localization of GABA synapses in the proximal dendrites shifted $E_{GABA}^{Thr}$ slightly towards more positive values (Fig 9E and 9F; S2 Table).

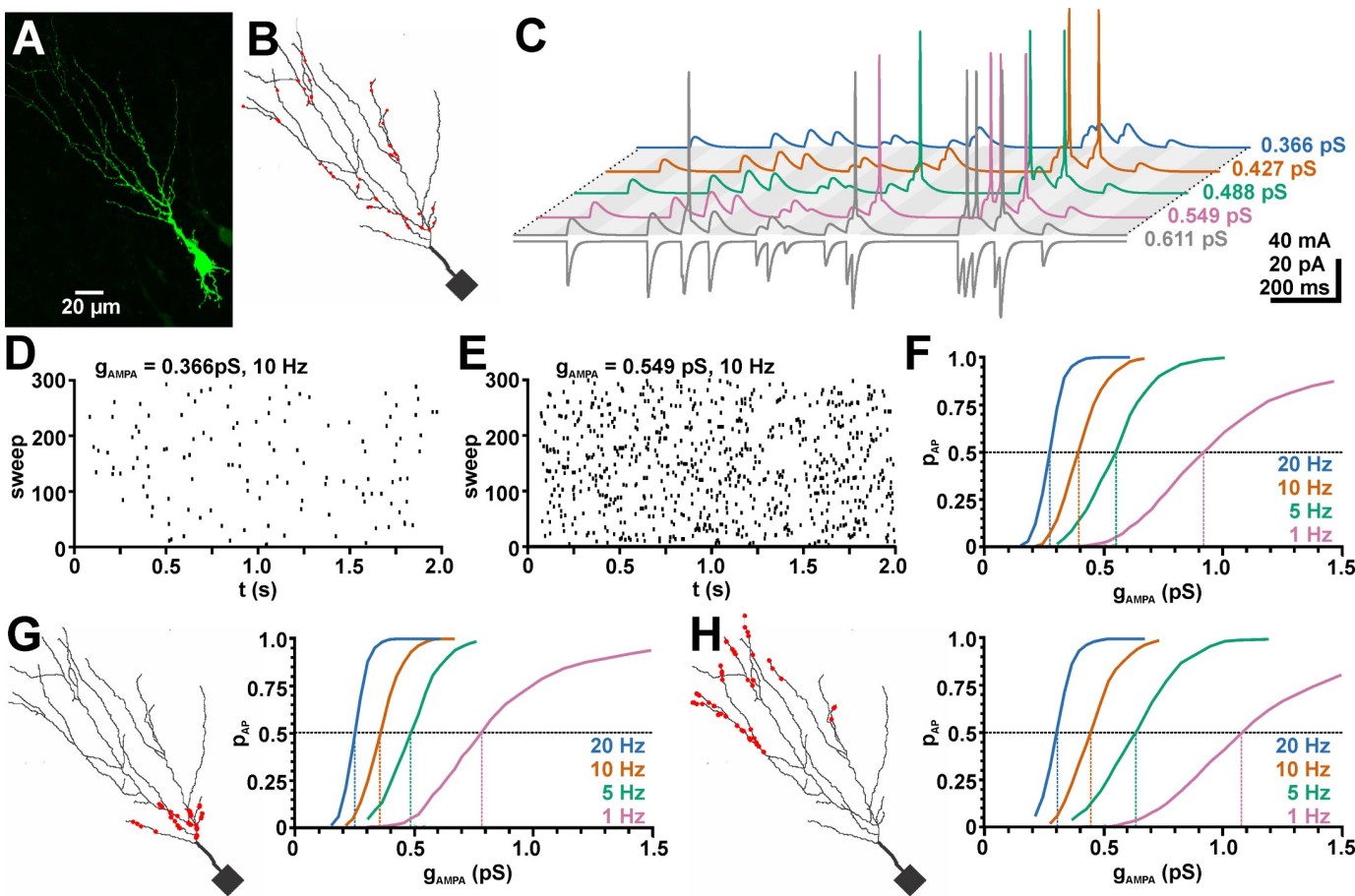

**Fig 7. Effect of AMPA mediated synaptic inputs on the excitability of a neuron with a dendritic topology derived from a reconstructed CA3 neuron.** A: Biocytin-FITC microfluorescence image of a CA3 pyramidal neuron. B. 2D model of this neuron. The red dots mark the location of randomly distributed AMPA synapses. C: The bottom trace illustrates the synaptic currents at a $g_{AMPA}$ of 0.611 pS and the color coded top traces the respective membrane responses at 5 different $g_{AMPA}$ for a stimulation frequency of 10 Hz. D, E: Raster plots depicting the occurrence of APs in 300 sweeps using random AMPA stimuli at a $g_{AMPA}$ of 0.366 pS and 0.549 pS, respectively. F: Probability for the occurrence of at least one AP ($p_{AP}$) at different $g_{AMPA}$. The dashed vertical lines indicate the $g_{AMPA}$ values at which $p_{AP}$ was 0.5. G: A slight excitatory shift was induced when AMPA synapses were restricted to the proximal dendrite. H: If AMPA synapses were restricted to the distal dendrite the spike probability function was shifted to larger $g_{AMPA}$ values. The $g_{AMPA}$ values obtained in these simulations were used for the determination of the GABAergic effects.

For higher frequencies this effect was even more pronounced when the AMPA synapses were restricted to distal dendrites. E.g. at 20 Hz stimulation with a gGABA of 2.27 nA $E_{GABA}^{Thr}$ amounted to −43 mV for evenly distributed AMPA and GABA synapses, which was shifted to −42.8 mV if the GABA synapses were restricted to proximal dendrites, and to −42.7 mV if in addition the AMPA synapses were restricted to the distal dendrites (S2 Table).

In summary, these results demonstrate that for uncorrelated high frequency inputs $E_{GABA}^{Thr}$ was close to $E_{Thr}^{ST}$, while it was more negative for lower frequencies. A slight but systematical $E_{GABA}^{Thr}$ shift in negative direction was observed for distal GABA inputs, while more proximal inputs brought $E_{GABA}^{Thr}$ even closer to $E_{Thr}^{ST}$. These observations suggest that for high frequency inputs GABA mediates a stable inhibitory effect as long as $E_{GABA}$ is negative to $E_{AP}^{Thr}$. For lower frequencies even less depolarized $E_{GABA}$ can mediate an excitatory effect. Although there is a systematic effect of the location of GABA receptors on $E_{GABA}^{Thr}$, only a small shift towards more stable inhibitory conditions for proximally located synapses can be deduced from these simulations.

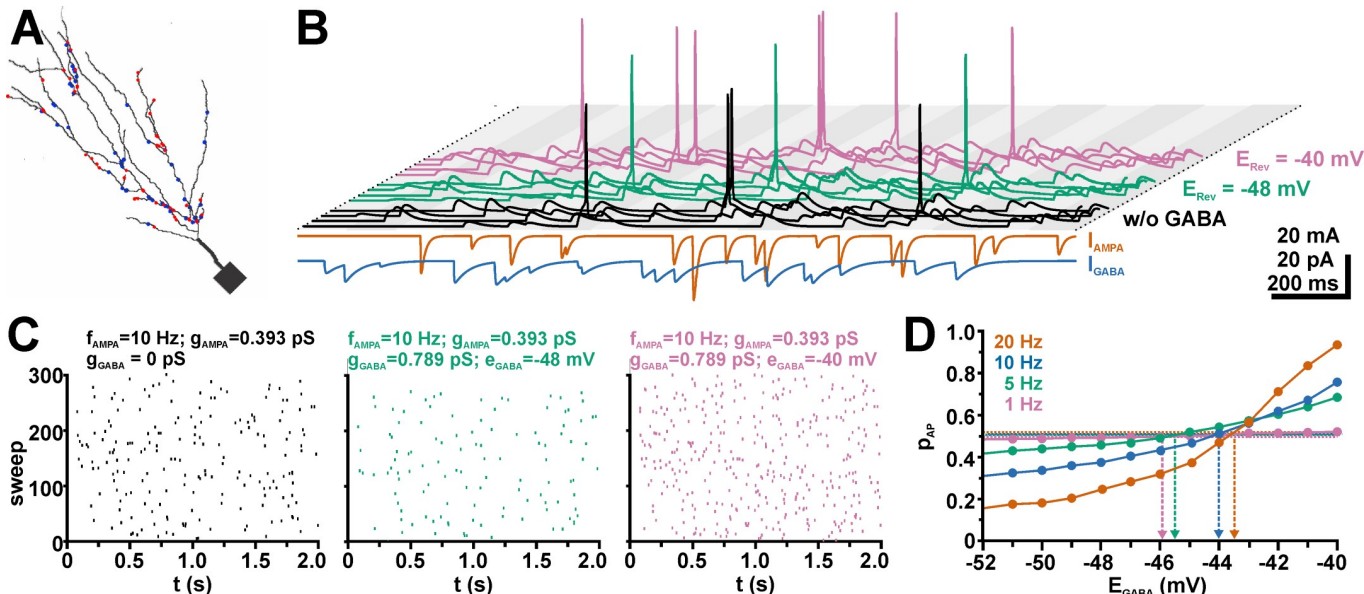

**Fig 8. Effect of GABAergic synaptic inputs on the excitability of a neuron with a dendritic topology derived from a reconstructed CA3 neuron.** A: 2D model of a CA3 neuron. The red dots mark the location of randomly distributed AMPA synapses and the blue dots that of randomly distributed GABA synapses. B: The bottom traces illustrate the synaptic glutamatergic (0.516 nS, orange) and GABAergic (0.789 nS; blue) currents. The top traces illustrate the voltage responses of 3 sweep obtained for only AMPA receptor stimulation (black) or with a GABAergic co-stimulation at $E_{GABA}$ of either −46 mV (orange) or −40 mV (blue). C: Raster plots depicting the occurrence of APs in 300 sweeps using only AMPA receptor stimulation (left panel) or GABA co-stimulation at $E_{GABA}$ of −46 mV (middle panel) or −40 mV (right panel). Note the inhibitory effect of GABA co-stimulation at $E_{GABA}$ of −46 mV and the excitatory effect at −40 mV. D: AP probability ($p_{AP}$) determined with frequencies of 1 Hz, 5 Hz, 10 Hz, and 20 Hz for a $g_{GABA}$ of 0.789 nS at different $E_{GABA}$. Note the sigmoidal dependency and that $p_{AP}$ reverses at $E_{GABA}$ between −43.5 mV and −45.9 mV.

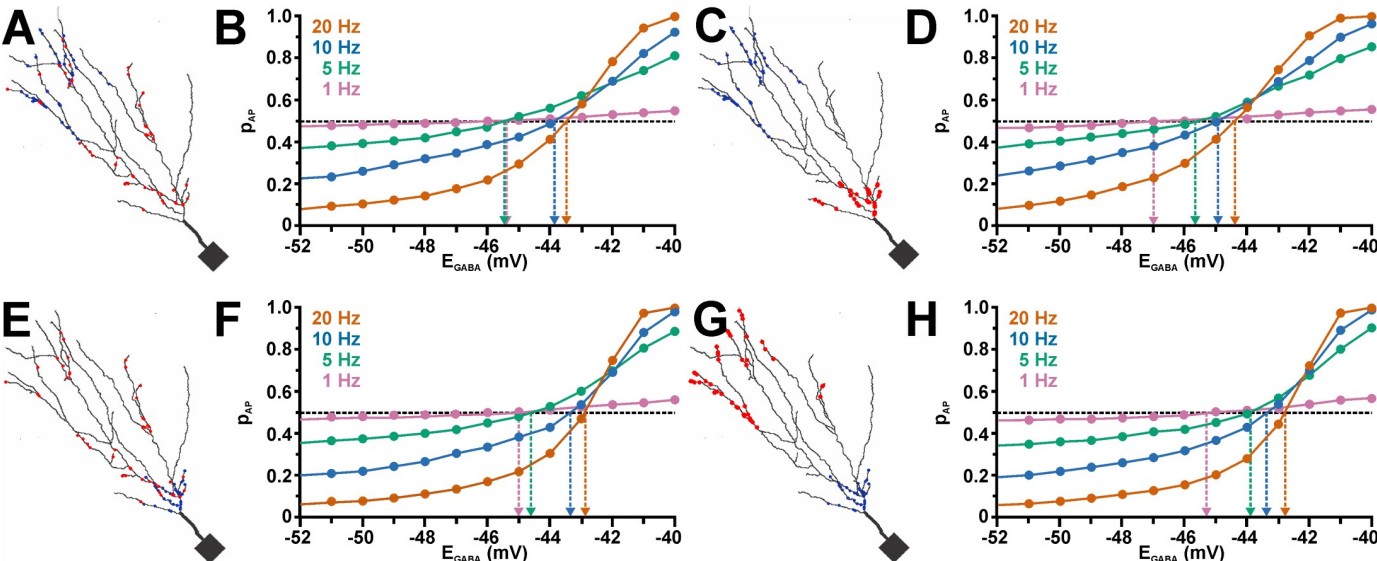

**Fig 9. Effect of the site of GABAergic synaptic inputs on $E_{GABA}^{Thr}$.** A: 2D model illustrating the random distribution of AMPA synapses (red dots), while the GABA synapses (blue dots) were restricted to distal dendrites. B: $p_{AP}$ vs. $E_{GABA}$ plot for a $g_{GABA}$ of 2.27 pS at 1 Hz, 5 Hz, 10 Hz, and 20 Hz, as identified by the colors. Note that the curve was comparable to the results obtained with a random distribution of GABA synapses (Fig 8D). C, D: As in A and B but for proximal AMPA synapses and distal GABA synapses. Note that $E_{GABA}^{Thr}$ was systematically shifted to negative values. E, F: As in A and B but for globally distributed AMPA synapses and proximal GABA synapses. Under this condition a slight positive shift in $E_{GABA}^{Thr}$ was observed for higher frequencies. G, H: As in A and B but for distal AMPA synapses and proximal GABA synapses. Note that the $E_{GABA}^{Thr}$ values were comparable to F.

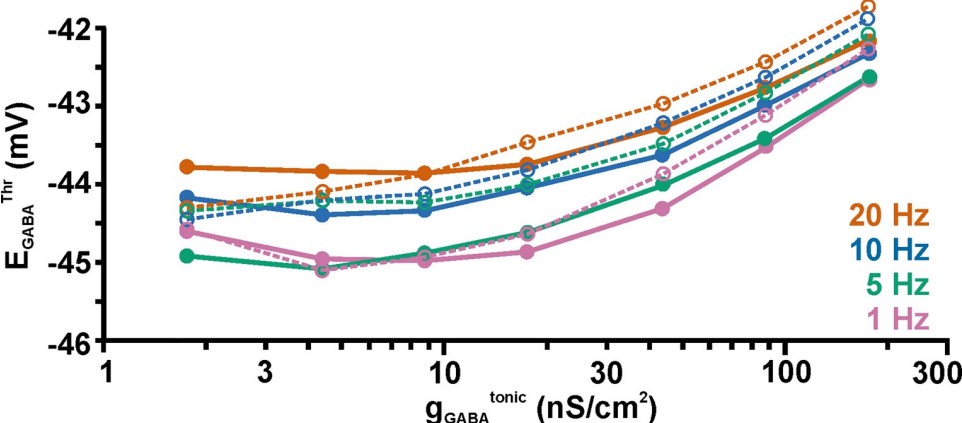

**Fig 10. Effect of tonic GABAergic conductances on $E_{GABA}^{Thr}$ in a neuron with a dendritic topology derived from a reconstructed CA3 neuron.** The graph displays $E_{GABA}^{Thr}$ values obtained for different $g_{GABA}^{tonic}$ and $f_{AMPA}$. Dashed lines/open symbols indicate $E_{GABA}^{Thr}$ under conditions when AMPA receptors were restricted to the distal dendrites. Note that at physiological $g_{GABA}^{tonic}$ values of 8.75 nS/cm$^2$, $E_{GABA}^{Thr}$ amounted to ca. -45 mV and was shifted to positive values with increasing frequencies. Augmenting $g_{GABA}^{tonic}$ systematically shifted $E_{GABA}^{Thr}$ towards $E_{AP}^{Thr}$. Note that the $E_{GABA}^{Th}$ values were slightly shifted to more positive values when the synapses were restricted to the distal compartment, however, this effect was negligible for physiological $g_{GABA}^{tonic}$. The intersection between the 1 Hz and 5 Hz curves at low $g_{GABA}^{tonic}$ was most probably caused by inaccuracies with the exact determination of changes in the spike probability induced by such small conductance shifts.

Finally, we also simulated the effect of tonic GABAergic currents on $E_{GABA}^{Thr}$ with this more realistic dendritic topology. For this purpose, we stimulated randomly distributed AMPA synapses at random time points using different frequencies between 1 Hz and 20 Hz, while simulating evenly distributed tonic GABAergic currents with different conductance densities $g_{GABA}^{tonic}$. The effect on $E_{GABA}^{Thr}$ was again quantified from the $E_{GABA}$ value at which the shift in the $p_{AP}$ curve upon random AMPA inputs changes from inhibition ($< 0.5$) to excitation ($> 0.5$). This simulation revealed that for the physiologically determined $g_{GABA}^{tonic}$ value of 8.75 nS/cm$^2$ [53], $E_{GABA}^{Thr}$ was at $-45$ mV for 1 Hz stimulation frequency and was shifted in depolarizing direction with increasing frequency of AMPA inputs (Fig 10, S3 Table). At higher $g_{GABA}^{tonic}$, $E_{GABA}^{Thr}$ was systematic shifted to positive values, approximating $E_{AP}^{Thr}$ (Fig 10). For simulations in which the AMPA receptors were restricted to the distal dendrites comparable results were observed (Fig 10). In summary, these results suggest that under consideration of a more realistic dendritic compartment and random AMPA inputs, tonic GABAergic currents reliably mediate an inhibitory effect as long as $E_{GABA}$ is less than ca. 1 mV depolarized to $E_{AP}^{Thr}$.

## 3. Discussion

Experimental findings indicate that $[Cl^-]_i$ and $[HCO_3^-]_i$ are dynamically shifted during early brain development, upon massive GABAergic activity and after pathophysiological insults [10,15,76]. Thus it became evident that GABA can have depolarizing actions [8,13] and this raised the question under which conditions the activation of GABA$_A$ receptors can mediate an excitatory effect. Theoretical considerations suggested that GABA$_A$ receptor activation permits an inhibitory effect as long as $E_{GABA}$ was below $E_{Thr}^{AP}$ [35,36]. However, this consideration just reflects a quasi one-dimensional situation and ignores the temporal and spatial components of GABAergic membrane responses as well as the restriction imposed by the passive membrane properties within more complex neuronal topologies [38–40]. Because the exact

role of GABA on the excitation/inhibition threshold is therefore hard to predict from such theoretical assumptions, we performed a detailed in-silico study using primary a simple neuronal topology and distinct spatiotemporal relations between GABAergic and glutamatergic inputs to evaluate at which $E_{GABA}$ values the net GABA effect switches from inhibitory to excitatory. In these simulations we were able to demonstrate that (i) for GABAergic synapses located close to the AP initiation zone (AIP) the difference between $E_{GABA}$ and $E_{AP}^{Thr}$ indeed reliably predicts whether GABA has an excitatory or inhibitory action. (ii) The threshold GABA reversal potential ($E_{GABA}^{Thr}$) was in this case close to the $E_{AP}^{Thr}$ defined by the maximal subthreshold current injection ($E_{Thr}^{ST}$). (iii) $E_{GABA}^{Thr}$ was systematically shifted to positive values with increasing distance between the GABA synapse and the AIP. (iv) An excitatory effect of GABA inputs on synchronous AMPA mediated inputs was observed when $E_{GABA}$ was above −44.9 mV, and thus consistently hyperpolarized to $E_{AP}^{Thr}$. (v) $E_{GABA}^{Thr}$ critically depends on the temporal relation between GABA and AMPA inputs, with a striking excitatory effect on AMPA-mediated inputs appearing after the GABA input. (vi) The spatial relation between GABAergic and AMPA-mediated inputs critically influences $E_{GABA}^{Thr}$, with $E_{GABA}^{Thr}$ systematically being shifted to values negative to $E_{AP}^{Thr}$ for AMPA synapses located proximally to the GABA input. (vii) The $E_{GABA}^{Thr}$ values for GABA synapses in the axon initial compartment were comparable to somatic GABA synapses. (viii) For tonic GABAergic conductances, $E_{GABA}^{Thr}$ was systematically negative to $E_{AP}^{Thr}$ over a wide range of $g_{GABA}^{tonic}$ values spanning the physiological range. (ix) Simulations using a neuron model with a realistic dendritic compartment revealed that $E_{AP}^{Thr}$ was only for high frequency inputs close to $E_{AP}^{Thr}$, but was slightly shifted to hyperpolarized values with lower frequencies and a more distal localization of GABA synapses. In summary, these results demonstrate that only for very restricted conditions the GABAergic effects switch from excitation to inhibition when $E_{GABA}$ was at $E_{AP}^{Thr}$. Under several physiologically relevant conditions, $E_{GABA}^{Thr}$ was negative to $E_{AP}^{Thr}$, suggesting that GABA can mediate excitatory effects already under these conditions.

It is important to note that in the present study we considered only $E_{GABA}$ as the relevant parameter, which in reality depends not only on $[Cl^-]_i$ but also on $[HCO_3^-]_i$ [6]. We have chosen this approach to (i) ease the computational load, (ii) because the consideration of two independent variables makes the interpretation of the results more complicated, and (iii) because the relative $HCO_3^-$ conductance of $GABA_A$ receptors differs between distinct neuronal subpopulations [6,77,78]. Differences in intracellular fixed charges can also slightly influence the relation between $[Cl^-]_i$, $E_{Cl}$ and the GABAergic driving force [79,80]. In addition, we did not consider that functionally relevant somato-dendritic $[Cl^-]_i$ gradients exists in neurons [11,81] and that GABAergic synaptic activity, alone or correlated to glutamatergic inputs, considerably alters $E_{GABA}$ [49,53,75,76,82–84]. All of these properties will complicate the prediction of GABAergic response direction, but for any interpretation of the functional consequences of temporal and spatially dynamic $[Cl^-]_i$ (and $[HCO_3^-]_i$) gradients, it will be necessary to obtain a major framework to understand how the GABAergic response direction depends on the relation between $E_{GABA}$, $E_{AP}^{Thr}$ and spatiotemporal synaptic properties. However, to ease the interpretation of the $E_{GABA}^{Thr}$ values, we estimated the corresponding $[Cl^-]_i$ using the Goldman equation and realistic parameters for hippocampal neurons (see materials and methods [49]). Using these parameters an $E_{GABA}$ of −50.5 mV corresponds to a $[Cl^-]_i$ of 14.5 mM, an $E_{GABA}$ of −42.8 mV (i.e. $E_{AP}^{Thr}$) to 21.7 mM, the $E_{GABA}^{Thr}$ of −44.7 mV observed for simultaneous GABA/AMPA synaptic inputs corresponds to 19.7 mM, and the $E_{GABA}^{Thr}$ of −47.5 mV observed for moderate tonic GABA inputs corresponds to a $[Cl^-]_i$ of 17.1 mM.

Previous studies reported for early postnatal cortical plate neurons an $E_{GABA}$ between ca. −40 mV [19,85] and −50 mV [86]. Thus the $E_{GABA}^{Thr}$ between −42.8 mV and −50.5 mV,

determined in the present study for the distinct conditions, is in the range of the experimentally measured $E_{GABA}$ values, indicating that GABA may indeed mediate excitatory as well as inhibitory effects in the immature neocortex. This suggestion is in line with studies reporting both, excitatory and inhibitory GABAergic effects in the immature brain [48,87]. In contrast, in immature hippocampal neurons an $E_{GABA}$ of ca. −55 mV [88] has been reported, which is clearly below the $E_{GABA}^{Thr}$ values determined in our study, suggesting a stable inhibitory GABAergic action in this brain region. However, several reports indicate that GABA can mediate excitation in immature hippocampal neurons [8,46,88], which is obviously in contrast to this suggestion. On the other hand, detailed analysis using minimally invasive recording methods indicate that $E_m$ and $E_{Thr}^{AP}$ are probably substantially more negative than observed with conventional whole-cell or gramicidin-perforated recording techniques, reporting an $E_m$ of −77 mV and an $E_{Thr}^{dVdt}$ of −46 mV [44,89]. Thus under physiological conditions both, $E_m$ and $E_{Thr}^{AP}$ are more negative than the values used for our simulation, supporting that a stable, inward directed GABAergic driving force during the first postnatal week can indeed exist [88]. However, all main findings of our in-silico study can directly be transferred to the more realistic conditions determined in the latter study, by applying a linear shift in the absolute values for $E_{GABA}^{Thr}$ and $E_{AP}^{ST}$ from the used parameters derived from conventional whole-cell recordings in immature hippocampal slices [49] to the more negative values suggested by Valeeva et al. [44].

Also of note is the observation that the threshold conductance $g_{GABA}^{Thr}$ determined for the simulation of only GABAergic synapses is orders of magnitude above physiological values for moderate GABAergic inputs of ca. 1 nS [49,53,54] when $E_{GABA}$ is approaching $E_{GABA}^{Thr}$. However, please consider that these conditions require that $E_{GABA}$ was only very slightly above $E_{GABA}^{Thr}$ and thus each GABAergic synapse contributed only a negligible depolarizing drive. In fact, from these simulations we could estimate that for an $E_{GABA}$ of ca. 0.5 mV positive to $E_{AP}^{ST}$ about 20–30 single synaptic inputs were required for a direct suprathreshold response, which is in the range of the observed number of correlated GABAergic inputs during a GDP, an excitatory transient network event depending on excitatory GABAergic synapses [8,49].

The first major result of this in-silico study was the observation that $E_{GABA}^{Thr}$ determined for the GABAergic effect on AMPA-mediated inputs was in many cases considerably negative to $E_{AP}^{Thr}$, in contrast to the initial theoretical consideration [35,36]. In our experiment we were also able to provide a mechanistic explanation for this observation. First, by using a current-clamp approach we could replicate that the GABAergic depolarization, when isolated from the GABAergic conductance shift, acted excitatory whenever the peak GABAergic depolarization was positive to the RMP, resulting in an $E_{GABA}^{Thr}$ of −50.5 mV. This stringent excitatory effect can be easily explained by the fact that in the absence of conductance changes each depolarization brings $E_m$ closer to $E_{AP}^{Thr}$. Next, we could demonstrate, by providing AMPA-inputs with a defined advance or delay to the GABAergic inputs, a clear bimodal effect of depolarizing GABA responses. In all cases in which the AMPA inputs preceded the GABA input $E_{GABA}^{Thr}$ was close to $E_{AP}^{Thr}$ (Fig 4H). Under this condition the AP initiation was under the control of the subsequent GABAergic conductance shift. And under this condition, the $GABA_A$ receptor will mediate an inward current, corresponding to a putative excitatory effect, as long as $E_{GABA}$ was positive to $E_m$, Thereby, an excitatory effect was induced only if $E_{GABA}$ was above $E_{AP}^{Thr}$. However, if the AMPA-mediated inputs occurred after the GABAergic inputs, $E_{GABA}^{Thr}$ was systematically shifted to more negative values approximating the RMP of −50.5 mV. This effect can be attributed to the fact that the GABAergic depolarization outlasts the GABAergic conductance shift. Thus, under these conditions the depolarization progressively dominates the effect of GABA, leading to a gradual shift in $E_{GABA}^{Thr}$ towards more negative potentials. If the GABAergic conductance can be neglected, each depolarizing shift, i.e.

each membrane change depolarized to RMP, contributed to the excitation, leading again to an $E_{GABA}^{Thr}$ of −50.5 mV. The impact of the temporal profile of GABAergic conductance change vs. GABAergic depolarization on neuronal excitability has already been experimentally addressed in hypothalamic [40] and neocortical [41] neurons, where within the same neuron the initial phase of a GABA response prevented AP initiation, whereas at later time points of the GABAergic responses AP initiation was facilitated. Despite this clear latency-dependent effect, the reciprocal actions of a depolarization-induced facilitation and a conductance-induced shunting inhibition can also explain why $E_{GABA}^{Thr}$ for synaptic inputs was neither at RMP, which would be the case if only the membrane potential shift was relevant, nor at $E_{Thr}^{AP}$, which would be the case if $E_m$ was only dependent on the actual GABAergic conductance. Note in this respect that the passive $E_{GABA}^{Thr}$ of −50.5 mV corresponds to an estimated $[Cl^-]_i$ of 14.4 mM.

In immature neurons, with their slow membrane time constants [63,90], the membrane responses are most probably prone to outlast the membrane conductance for both glutamatergic and GABAergic synaptic inputs. On the other hand, this effect of a prolonged membrane time constant in immature neurons may be partially compensated by the fact, that immature synaptic GABAergic currents show significantly longer decay time constants [63], thereby prolonging the interval in which the shunting inhibitory effects contributes to $E_{GABA}^{Thr}$. Another important functional consequence of our results is that the timing between GABAergic and glutamatergic inputs critically determines $E_{GABA}^{Thr}$. In this respect classical feedforward as well as recurrent inhibition, with its short latency to excitatory inputs [91], will impose a rather strict inhibition even at depolarizing GABAergic conditions as long as $E_{GABA}$ is maintained below $E_{Thr}^{AP}$. Thus this kind of inhibitory control would be rather stable upon activity dependent shifts in $E_{GABA}$ [49,76,82,83,92]. On the other hand, for GABAergic inputs that are not temporally correlated with the excitatory inputs, e.g. during blanket inhibition, it must be considered that $E_{GABA}^{Thr}$ can be negative to $E_{AP}^{Thr}$, and thus may mediate a less stable inhibition that is more sensitive to ionic plasticity.

The second major result of this in-silico study was the observation, that the spatial relation between GABAergic and AMPA inputs also critically affects $E_{GABA}^{Thr}$. As expected, our simulation revealed that the inhibitory effect, as quantified by $\Delta g_{AMPA}^{Thr}$, of proximal GABAergic synapses are stronger than that of distally located ones. The $\Delta g_{AMPA}^{Thr}$ values were substantially larger for AMPA synapses located distally to the GABA synapse, indicating that a GABA input can shunt EPSPs from distal synapses, as suggested from in-vitro and in-silico experiments [41]. For proximally located GABA synapses we could observe that $E_{GABA}$ showed only little dependency on the location of the AMPA-mediated inputs. In these cases, $E_{GABA}^{Rev}$ amounted to ca. −46 mV, suggesting that both, shunting and depolarizing effects contribute to the impact of GABA on the excitability. In contrast, we observed for distally located GABA synapses a strong dependency of $E_{GABA}^{Thr}$ on the location of AMPA-mediated inputs. For such distal GABA synapse locations a negative $E_{GABA}^{Thr}$ close to −50 mV was observed at proximal AMPA synapses, which reflects the fact that with this configuration only the electrotonically propagating GABAergic depolarization has an effective influence with the AMPA-mediated depolarization, while the GABAergic conductance shift acts more locally. For co-localized GABA and AMPA synapses at the distal end of the dendrite $E_{GABA}^{Thr}$ approximated $E_{AP}^{Thr}$ at ca. −43 mV, indicating that here the effect of GABA was mediated mainly by membrane shunting. Intriguingly the "slope" of $E_{GABA}^{Thr}$ was steeper for AMPA synapses in the dendritic segment proximal to the GABA synapse. The slope became shallower for the segment distal from the GABA synapse. This observation indicates that for all AMPA synapses distal to the GABA synapse a substantial fraction of the synaptic currents were shunted by the GABAergic conductance before they can affect AP initiation in the soma. In contrast, for all

AMPA synapses located proximal to the GABA synapse the shunting effect was diminished with increasing distance between both synapses, whereas the electrotonically propagating depolarization maintained a more stable excitatory influence and thereby shifted $E_{GABA}^{Thr}$ towards the RMP. Thus the results of our experiments suggest an additional mechanism that contribute the putative excitatory GABAergic effect of dendritic GABA inputs [41], in addition to the existence of stable or dynamic somato-dendritic $[Cl^-]_i$ gradients [93,94].

These in-silico observations indicate that perisomatic inhibition, which is the dominant form for the classical feedback and feedforward inhibition mediated by parvalbumin-positive interneurons [95,96], can maintain a stable inhibitory effect regardless of the site of glutamatergic inputs and ionic plasticity. On the other hand, the impact of GABAergic synapses located in the dendritic periphery, e.g. by the hippocampal O-LM interneurons [97] or neocortical Martinotti interneurons [98], will critically depend on the location of the depolarizing GABAergic inputs and can putatively mediate an excitatory impact on AMPA synapses close to the soma at slightly depolarizing $E_{GABA}$.

A specific physiological function has been suggested for the synapses of Chandelier cells on the axon initial segment, as it has been reported that these synapses maintain a depolarizing, putatively inhibitory action [70,71]. However, other reports suggest that GABA receptors in the axon initial segment still mediate an inhibitory effect due to a depolarized shift in the AP threshold [99]. Our results indicate that a GABAergic synapse at the axon initial segment mediated an inhibitory action on somatic glutamatergic inputs as long as $E_{GABA}$ was slightly below the $E_{AP}^{Thr}$, whereas $E_{GABA}^{Thr}$ was shifted to negative values for glutamatergic inputs located in the dendrite. However, the $E_{GABA}^{Thr}$ estimated from these simulations only marginally differ from values obtained with a somatic GABAergic synapse and the same organization of glutamatergic inputs. In summary, these results indicate that the GABA synapse in the axon initial segment does not represent a specific synapse when the dependency between $E_{GABA}$ and excitability control was considered. But independent of this conclusion, the specific properties of $[Cl^-]_i$ homeostasis in the axon initial segment [70,71] as well as the impact of this synapse on the AP threshold [99] will still infer rather specific implications of this synapse in regulating neuronal spike output.

For the present simulation we used in the ball and stick model and the reconstructed dendritic topology only passive dendritic membranes. However, in reality dendrites are equipped with a collection of ion channels that underlie non-linear integration and enable active information processing within this compartment [100,101]. It is generally considered that mainly a supralinear integration occurs in active dendrites [100]. The apparent reduction in the dendritic filtering under this supralinear integration would reduce the slope of the $E_{GABA}^{Thr}$ gradient between the proximal and distal end of the dendrite (Fig 5). In the dendrite also anterograde APs can be initiated. In hippocampal pyramidal neurons they are generated at higher stimulus intensities than somatic APs [102] and would thus not interfere with the determination of $E_{GABA}^{Thr}$. In other case dendritic APs are evoked at lower stimulation thresholds [103], which will most probably led to the situation that $E_{GABA}^{Thr}$ will approximate the $E_{AP}^{Thr}$ of the dendritic AP.

In addition, our results indicate that for small to moderate tonic GABAergic conductance $E_{GABA}^{Thr}$ was systematically more negative than $E_{AP}^{Thr}$, which suggests that even at rather moderate depolarizations tonic GABAergic currents can mediate an excitatory effect. Only at higher $g_{GABA}^{tonic}$ the $E_{GABA}^{Thr}$ approaches $E_{AP}^{Thr}$. The results of this simulation replicate the findings of a previous in-vitro study, that demonstrated excitatory effects of depolarizing tonic GABAergic responses at low conductances, whereas at higher conductances a stable inhibition was imposed [104]. Our results are also in line with the excitatory effects of extrasynaptic $GABA_A$ receptors in the immature hippocampus [45]. In our simulations $E_{GABA}^{Thr}$ remained

stable at about −48.3 mV for $g_{GABA}^{tonic}$ smaller than ca. $10^{-2}$ nS/cm$^2$, which is close to the passive membrane conductance $g_{pas}$ of 0.0128 nS/cm$^2$. We assume that below this value the shunting effects caused by $g_{GABA}^{tonic}$ were negligible to the background conductance $g_{pas}$ and thus did not considerably contribute to the shunting of EPSCs. Only if $g_{GABA}^{tonic}$ exceeded $g_{pas}$ a relevant additional inhibitory component was imposed by the GABAergic conductances and thus $E_{GABA}^{Thr}$ converged towards $E_{AP}^{Thr}$.

The results from the simulation in a neuron with a more realistic dendritic morphology are mainly congruent to the results with the simplified dendritic geometry. At high frequencies, which resulted in a high probability that AMPA inputs occurred during the GABAergic conductance shift, we observed that $E_{GABA}^{Thr}$ was close to $E_{AP}^{Thr}$. Thus under physiological conditions in desynchronized states [105] or during synchronized activity states in the immature brain [8,49], which are characterized by a high frequency of both GABAergic and glutamatergic inputs, GABA mediates a stable inhibitory effect as long as $E_{GABA}$ was slightly below $E_{AP}^{Thr}$. As a consequence, such kind of GABAergic inhibition is less prone to activity-dependent $[Cl^-]_i$ increases [76,82,106]. For low frequencies of unsynchronized inputs, which resulted in a high probability that the AMPA inputs happen during the late, depolarization-dominated phase of a GABA response, $E_{GABA}^{Thr}$ is less positive, indicating that under such conditions excitatory GABAergic effects can happen at lower $[Cl^-]_i$. In addition, our in-silico experiments with the more realistic dendritic morphology also indicates that proximally located GABAergic synapses, which represents an important class of GABA synapses mediating feedforward inhibition [68], are even more resistant to $[Cl^-]_i$ alterations with an $E_{GABA}^{Thr}$ around $E_{AP}^{Thr}$. Thus, this type of synapse, located in a dendritic compartment that due to its dimensions is already less prone to dynamic $[Cl^-]_i$ changes, can maintain an inhibitory action already at rather high synaptic activity levels. On the other hand, our simulations also revealed that for uncorrelated GABA and AMPA inputs in a frequency range between 1 and 20 Hz $E_{GABA}^{Thr}$ was under all condition above -46 mV, indicating that a substantially high $[Cl^-]_i$ of about 18.5 mM would be required to mediate an excitatory effect. Similar conclusion could be drawn for the influence of tonic GABAergic receptors, where the $E_{GABA}^{Thr}$ were at about -45 mV, corresponding to a $[Cl^-]_i$ of about 19.5 mM. We assume that this discrepancy in the $E_{GABA}^{Thr}$ values between the reduced and the realistic dendritic model may be caused by the fact that in the approach we used for the model with a simple dendritic morphology (only a single AMPA synapse), we did not consider the effect of tonic GABAergic conductances on the temporal summation of glutamatergic postsynaptic potentials. In summary, the experiments with the more realistic topology indicate that an effect of spatial and temporal relation between AMPA and GABA inputs on $E_{GABA}^{Thr}$ exists, but is in the range of few mV. However, from our experiments with random inputs it cannot be excluded that for specific conditions, with remote and temporally separated synaptic inputs, $E_{GABA}^{Thr}$ may also be substantially more hyperpolarized.

Another conclusion that could be drawn from our study is that some attention should be taken to the method used to detect the AP threshold. Obviously there is, despite the frequent use of this descriptive parameter, no consensus on the definition of AP threshold [43]. Therefore, we used in this in-silico study four different, established methods for $E_{AP}^{Thr}$ detection. Our in-silico experiments demonstrated that the AP threshold value determined from a fixed threshold of dV/dt [44,51], from the first positive peak in d$^3$V/dt$^3$ [52], and from linear regression of the AP upstroke [37] were comparable at potentials of ca. −34 mV to ca. −38 mV. In contrast, substantially negative values of −42.8 mV were determined if $E_{AP}^{Thr}$ was defined as the maximal potential that did not result in AP triggering ($E_{Thr}^{ST}$). The difference in the results of these methods can be easily explained by the fact that $E_{Thr}^{ST}$ represents a quasi-stationary value (dV/dt close to 0) that is just insufficient to trigger the entry to the Hodgkin cycle. On

the other hand, the other three $E_{AP}^{Thr}$ values represent distinct states during the dynamic events in the initial AP phase. The fact that in our simulations $E_{GABA}^{Thr}$ for only GABAergic inputs indeed approximated $E_{Thr}^{ST}$ can be related to the fact that the excitation threshold for GABAergic inputs was also determined under quasi-stationary conditions. For the influence of GABA on synaptic AMPA-mediated inputs the excitation threshold was determined in the interval between the onset of the GABA inputs and the duration at which 63% of the peak depolarization was obtained. Thus, for the relevant traces that distinguished between sub-threshold and suprathreshold AMPA inputs, dV/dt was considerable small and thus the AP threshold was also determined under quasi stationary conditions. Under physiological conditions random fluctuation in $E_m$ will most probably limit the difference between $E_{Thr}^{dVdt}$, $E_{Thr}^{d3}$, $E_{Thr}^{IS}$, and $E_{Thr}^{ST}$. In any way, while addition of membrane noise to the in-silico models and/or a different methodological definition of the excitation threshold for GABA- and AMPA-mediated inputs would probably change the absolute values for $E_{GABA}^{Thr}$ and $E_{AP}^{Thr}$, it would not substantially interfere with the main observation of this study, that $E_{GABA}^{Thr}$ is for many physiologically relevant situations negative to $E_{AP}^{Thr}$.

In conclusion, this simulation indicates that, in addition to the influence of short-term and long-term ionic plasticity, the uneven distribution of $[Cl^-]_i$ gradients within individual cells and the effects of tonic and phasic inhibition [10,11,76,82], the observed spatial and temporal constraints on the $E_{GABA}$ to $E_{AP}^{Thr}$ relation imposes another level of complexity to the dynamic properties of GABAergic inhibition/excitation. While on one hand our results support the textbook knowledge that GABA mediates a stable inhibition as long as hyperpolarizing membrane responses are evoked (or $[Cl^-]_i$ is sufficiently low), on the other hand the altered $[Cl^-]_i$ homeostasis in early development and several neurological conditions like trauma, stroke or epilepsy [11,12,30,31], can impact the level of inhibitory control already upon moderate $[Cl^-]_i$ changes in a complex way.

## 4. Materials and methods

### 4.1 Ethics statement

All experiments were conducted in accordance with EU directive 86/609/EEC for the use of animals in research and the NIH Guide for the Care and Use of Laboratory Animals, and were approved by the local ethical committee (Landesuntersuchungsanstalt RLP, Koblenz, Germany). We made all efforts to minimize the number of animals and their suffering.

### 4.2. Electrophysiological procedures

**4.2.1. Slice preparation.** Newborn pups of postnatal days [P] 4–7 were obtained from time pregnant C57Bl/6 mice (Janvier Labs, Saint Berthevin, France) housed in the local animal facility at 12/12 day/night cycle and ad libitum access to food and water. The mouse pups were decapitated in deep enflurane (Ethrane, Abbot Laboratories, Wiesbaden, Germany) anaesthesia, their brains were quickly removed and immersed for 2–3 min in ice-cold standard artificial cerebrospinal fluid (ACSF, 125 mM NaCl, 25 mM NaHCO$_3$, 1.25 mM NaH$_2$PO$_5$, 1 mM MgCl$_2$, 2 mM CaCl$_2$, 2.5 mM KCl, 10 mM glucose, equilibrated with 95% O$_2$ / 5% CO$_2$, osmolarity 306 mOsm). Four hundred μm thick horizontal slices including the hippocampus were cut on a vibratome (Microm HM 650 V, Thermo Fischer Scientific, Schwerte, Germany) and subsequently stored in an incubation chamber filled with oxygenated ACSF at room temperature for at least 1h before they were transferred to the recording chamber.

**4.2.2 Patch-clamp recordings.** Whole-cell patch-clamp recordings were performed at 31 ± 1˚C in a submerged-type recording chamber attached to the fixed stage of a microscope (BX51 WI, Olympus). Pyramidal neurons in the stratum pyramidale of the CA3 region were

identified by their location and morphological appearance in infrared differential interference contrast image. Patch-pipettes (5–12 MΩ) were pulled from borosilicate glass capillaries (2.0 mm outside, 1.16 mm inside diameter, Science Products, Hofheim, Germany) on a vertical puller (PP-830, Narishige) and filled with the pipette solutions (86 mM K-gluconate, 44 mM KCl, 4 mM NaCl, 1 mM CaCl$_2$, 11 mM EGTA, 10 mM K-HEPES, 2 mM Mg2-ATP, 0.5 mM Na-GTP, pH adjusted to 7.4 with KOH and osmolarity to 306 mOsm with sucrose). In few experiments 40 mM KCl were replaced with 40 mM K-gluconate. Signals were recorded with a discontinuous voltage-clamp/current-clamp amplifier (SEC05L, NPI, Tamm, Germany), low-pass filtered at 3 kHz and stored and analyzed using an ITC-1600 AD/DA board (HEKA) and TIDA software. All voltages were corrected post-hoc for liquid junction potentials of -8 mV for a pipette [Cl$^-$] of 10 mM and -5 mV for 50 mM [20]. Input resistance and capacitance were determined from a series of hyperpolarizing current steps. Action potentials (AP) were induced by a series of depolarizing current steps. For averaging of AP wave forms the first AP from traces that showed a series of APs were used.

## 4.3. Compartmental modeling

The compartmental modeling was performed using the NEURON environment (neuron.yale. edu). The source code of all models and stimulation files used in the present paper can be found in ModelDB (http://modeldb.yale.edu/267142). For compartmental modelling we used either a simple ball (soma diameter = 43 μm) or a ball and stick model (soma with d = 43 μm, linear dendrite with L = 1000 μm, diameter 1 μm, and 301 segments). In both models a passive conductance ($g_{pas}$) with a density of 1.28*10$^{-5}$ nS/cm$^2$ and a reversal potential ($E_{pas}$) of −50.5 mV was distributed, which resulted for the ball-and-stick model in passive membrane properties that were comparable to the properties of recorded pyramidal CA3 neurons. Active membrane properties were in the majority of the experiments incorporated only in the somatic compartment. In one experiment we added to the ball and stick model an axon containing one initial segment (L = 10 μm, diameter = 0.2 μm) with active AP properties and remaining 10 segments (L = 10 μm, diameter = 0.2 μm) in which the Na$^+$ and K$^+$ peak conductivity was reduced by 10% [99]. In this experiment the GABA synapse was restricted to the axon initial segment.

Because it was not possible to generate a reasonable sharp AP onset with a standard Hodg-kin-Huxley (HH) model and since we are particularly interested in the AP threshold properties, we adapted a model developed by Naundorf et al. [50]. This model considered three different states for the Na$^+$ channels:

$$Na_o = open\ state$$

$$Na_c = closed\ state$$

$$Na_i = inactivated\ state$$

With mutual transitions between Na$_o$ and Na$_c$ as well as Na$_c$ and Na$_i$ and a mono-directional transition from Na$_o$ to Na$_i$. The rate functions $\alpha_A(V)$ for the transition Na$_c$➔Na$_o$ and $\alpha_{IC}(V)$ for the transition Na$_i$➔Na$_c$ are described by the functions:

$$\alpha_A(V_t) = \frac{Q_{10}}{\tau_{Naact}} \times \frac{G_{c \to o}^{Na}}{\left(1 + e^{\left(\frac{(V_{c \to o}^{Na} - V_t)}{k_{c \to o}^{Na}}\right)}\right)}\ and\ \alpha_{IC}(V_t) = \frac{Q_{10}}{\tau_{Naina}} \times \frac{G_{i \to c}^{Na}}{\left(1 + e^{\left(\frac{(V_{i \to c}^{Na} - V_t)}{k_{i \to c}^{Na}}\right)}\right)}.$$

The rate functions $\beta_A(V)$ for the transition $Na_c \rightarrow Na_o$ and $\beta_{IC}(V)$ for the transition $Na_c \rightarrow Na_i$ are described by the function:

$$\beta_A(V_t) = \frac{Q_{10}}{\tau_{Naact}} \times \frac{G^{Na}_{o \rightarrow c}}{\left(1 + e^{\left(\frac{(V_t - V^{Na}_{c \rightarrow o})}{k^{Na}_{c \rightarrow o}}\right)}\right)} \; and \; \beta_{IC}(V_t) = \frac{Q_{10}}{\tau_{Naina}} \times \frac{G^{Na}_{i \rightarrow c}}{\left(1 + e^{\left(\frac{(V_t - V^{Na}_{c \rightarrow c})}{k^{Ni}_{i \rightarrow c}}\right)}\right)}.$$

The voltage independent relaxation from $Na_o$ occurs with the rate constant $\tau_{Na}$.

The dynamic properties of the fraction of open $Na^+$ channels $O_{Na}$ and inactivated $Na^+$ channels $H_{Na}$ were described by the differential equations:

$$\dot{O}_{Na} = \alpha_A(v + cf_{Na}O_{Na})(1 - O_{Na} - H_{Na}) - \beta_A(v + cf_{Na}O_{Na})O_{Na} - \frac{O_{Na}}{\tau_{Na}}$$

$$\dot{H}_{Na} = \alpha_{IC}(v)(1 - H_{Na}) - \beta_{IC}(v)H_{Na} - \frac{H_{Na}}{\tau_{Na}}$$

The cooparativity factor $cf_{Na}$ was introduced by Naudorf et al. to account for the cooperative opening of $Na^+$ channels under realistic condition [50].

The actual $Na^+$ conductance $g_{Na}$ was given by the equation $g_{Na} = g_{Na}{}^{Max} O_{Na}$.

The $Na^+$ current $I_{Na}$ was calculated from $g_{Na}$ and the sodium equilibrium potential $e_{Na}$ according to Ohm's law:

$$I_{Na} = g_{Na}(v - e_{Na}).$$

In addition to the exclusive $Na^+$ current model published by Naundorf et al., we implemented a simple two state model for the delayed rectifier $K^+$ current to enable the simulation of action potentials. The $K_c \rightarrow K_o$ transition rate described by the equation:

$$\alpha^K_A(V_t) = \frac{Q_{10}}{\tau_{Kact}} \times \frac{G^K_{c \rightarrow o}}{\left(1 + e^{\left(\frac{(V^K_{c \rightarrow o} - V_t)}{k^K_{c \rightarrow o}}\right)}\right)}.$$

The $K_o \rightarrow K_c$ transition rate was described by the function

$$\beta^K_A(V_t) = \frac{Q_{10}}{\tau_{Kina}} \times \frac{G^K_{i \rightarrow c}}{\left(1 + e^{\left(\frac{(V_t - V^K_{i \rightarrow c})}{k^K_{i \rightarrow c}}\right)}\right)}.$$

In addition, a voltage independent relaxation from $K_o$ with the rate constant $\tau_K$. was considered.

The dynamic properties of the open fraction of $K^+$ channels ($O_K$) was described by the differential equations using the cooparativity factor $cf_K$:

$$\dot{O}_K = \alpha_A(v + cf_K O_{Na})(1 - O_K) - \beta_A(v + cf_K O_K)O_K - \frac{O_K}{\tau_K}.$$

The actual $K^+$ conductance $g_K$ was given by the equation $g_K = g_K{}^{Max}O_K$ and the $K^+$ current ($I_K$) was calculated by the equation $I_K = g_K(v - e_K)$.

All parameters were optimized by stepwise approximation to obtain a sufficient fit to the average experimentally determined AP trace, which was quantified by minimizing the root of the summarized squared errors according to the following error weight function:

$$Error = 10 \times \sqrt{\left(E_{Thr}^{d3}\right)^2} + 3 \times \sqrt{\left(v_{rise}^{max}\right)^2} + \sqrt{\left(v_{decay}^{max}\right)^2} + \sqrt{\left(d_{1/2}\right)^2} + \sqrt{\left(E_{AP}^{Peak}\right)^2}.$$

This error weight function was used with the rationale to put special emphasis for the fitting routine to the dynamic properties at $E_{AP}^{Thr}$. The used parameters are given in S4 Table.

AMPA synapses were modeled by an Exp2Syn point process using a reversal potential of -12 mV, a tau1 value of 0.1 ms and a tau2 value of 11 ms, in accordance with the experimentally determined value [49]. GABA synapses were modeled by an Exp2Syn point process using a tau1 value of 0.1 ms and a tau2 value of 37 ms, in accordance with the experimentally determined value [49]. The reversal potential of the GABAergic synapses was the main variable of interest in these simulations. For tonic GABAergic currents a constant membrane conductance was distributed over all membrane with conductance densities and reversal potentials as given in the results part [53].

For the determination of $g_{GABA}^{Thr}$ we used an iterative approach where $g_{GABA}$ was first increased by 1 nS steps until an AP was induced within an interval of 800 ms after the GABA input. Subsequently $g_{GABA}$ was decreased by 0.33 nS steps until the AP vanished, followed again by an increase in $g_{GABA}$ by 0.1 pS until the AP reappeared. This alternating sequence was repeated 6 times using a $g_{GABA}$ of 1/10 for each subsequent round. In these experiments $E_{AP}^{Thr}$ was defined as the peak voltage of the last subthreshold sweep.

A similar approach was also used to determine $g_{AMPA}^{Thr}$. Here $g_{AMPA}$ was initially increased by 0.01 nS steps until an AP was induced. The analysis interval was in all sweeps set to the interval between stimulus onset and the time point when the AMPA-mediated depolarization, determined in the absence of an AP mechanism, decreased to 63% of the peak amplitude. Subsequent $g_{AMPA}$ was decreased by 3.3 pS until the AP disappears, followed by 6 rounds of alternating increasing/decreasing $g_{AMPA}$ steps, with $g_{AMPA}$ step values decreasing by 1/10 for each round.

Due to this iterative approach 55 ± 2.2 sweeps (in n = 22 simulations) were required for each of the 1827 parameters tested to determine $g_{GABA}^{Thr}$ and 63.3 ± 0.8 sweeps (in n = 85 simulations) were required for each of the 29730 parameters tested to determine $E_{GABA}^{Thr}$. In consequence, between 37348 (Fig 4C) and 534907 (Fig 5B) sweeps are required to test a hypothesis.

To determine $E_{GABA}^{Thr}$ under more realistic conditions, we used a dendritic model derived from a reconstructed, biocytin-labeled CA3 pyramidal neuron (Fig 7A and 7B) [49]. This model consists of a soma (diameter 15 μm) and 56 dendrites, that contained between 2 and 193 segments, as adapted from the reconstruction. To determine $E_{GABA}^{Thr}$ in this model 2, 10, 20 or 40 GABA and AMPA synapses, for respective stimulation frequencies of 1, 5, 10 or 20 Hz, were randomly distributed across the dendritic compartment and each synapse was stimulated at a random time point during the 2 s stimulation interval. The value for $g_{AMPA}$ was set to values that corresponds to a spike probability (pAP) of 0.5, determined for each frequency from 999 single sweeps using the same random number sequence in the absence of GABAergic inputs. The $p_{AP}^{50}$ was calculated from a linear interpolation of the two values closest to a $p_{AP}$ of 50%. $E_{GABA}^{Thr}$ was determined from the $E_{GABA}$ values at which the $p_{AP}$ curve obtained in 999 sweeps reaches the $p_{AP}^{50}$ value determined with the same AMPA stimulation pattern in the absence of GABA.

The $[Cl^-]_i$ was estimated from the determined $E_{GABA}^{Thr}$ using the Goldman-Hodgkin-Katz equation as follows:

$$[Cl^-]_i = 10^{\frac{E_{GABA}}{60mV}}\left(P_{Cl}[Cl^-]_e + P_{HCO3}[HCO_3^-]_e\right) - P_{HCO3}[HCO_3^-]_i$$

using a $[Cl^-]_e$ of 133.5 mM, an extracellular $HCO_3^-$ concentration ($[HCO_3^-]_e$) of 24 mM, a $[HCO_3^-]_i$ of 14.1 mM, and a relative $HCO_3^-$ permeability ($P_{HCO}$) of 0.44 [49].

## Supporting information

**S1 Fig. Characterization of AP properties using different dt values for the simulation.** A: Simulated voltage traces using different dt as indicated in the plot. Note the slightly divergent AP shape at 0.05 ms, while at a dt of 0.5 ms oscillations occur. B: Rate of $E_m$ changes during an action potential. C: Typical $E_{AP}^{Thr}$ values determined with 3 different algorithms on the traces obtained at different dt. Note that all $E_{Thr}^{IS}$, $E_{Thr}^{dV/dt}$ and $E_{Thr}^{d3}$ remained stable for a dt $\leq$ 0.025 ms.
(TIF)

**S2 Fig. Effect of a GABA synapse on the excitability of a ball and stick neuron with an added axon.** AP mechanisms were restricted to the axon and the GABA synapse was located at the somatic end of the axon (axon initial segment). A: Plot of $\Delta g_{AMPA}^{Thr}$ versus $E_{GABA}$ at different $g_{GABA}$ values for an AMPA synapse located at the soma. B: Plot of $E_{GABA}^{Thr}$ at different $g_{GABA}$ for somatic AMPA receptors. Note that $E_{GABA}^{Thr}$ was ca. -44.2 mV for physiological $g_{GABA}$ and was shifted towards $E_{Thr}^{IS}$ at higher $g_{GABA}$. C: Plot of $E_{GABA}^{Thr}$ at different $g_{GABA}$ for dendritic AMPA receptors located at 25%, 50% and 57% of the dendrite, as indicated by the color code. The grey trace represents the somatic AMPA stimulation. Note that $E_{GABA}^{Thr}$ was systematically shifted towards lower values with more distant dendritic location.
(TIF)

**S1 Table. List of $g_{AMPA}$ values required to reach 50% probability for spike initiation at different distributions and frequencies of AMPA inputs.**
(DOCX)

**S2 Table. Summary of EGABA$^{Thr}$ values determined in the experiments with the reconstructed dendritic topology using synaptic GABAergic currents.**
(DOCX)

**S3 Table. Summary of EGABA$^{Thr}$ values determined in the experiments with the reconstructed dendritic topology using tonic GABAergic currents.**
(DOCX)

**S4 Table. Parameters used for the modified Naundorf model.**
(DOCX)

## Author Contributions

**Conceptualization:** Werner Kilb.

**Formal analysis:** Aniello Lombardi, Werner Kilb.

**Investigation:** Aniello Lombardi.

**Methodology:** Werner Kilb.

**Writing – original draft:** Aniello Lombardi, Heiko J. Luhmann, Werner Kilb.

**Writing – review & editing:** Aniello Lombardi, Heiko J. Luhmann, Werner Kilb.

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
