## [Decision Letter · Decision Letter 0]

23 Jul 2021

Dear Dr. Kilb,

Thank you very much for submitting your manuscript "Modelling the spatial and temporal constrains of the GABAergic influence on neuronal excitability" for consideration at PLOS Computational Biology.

As with all papers reviewed by the journal, your manuscript was reviewed by members of the editorial board and by several independent reviewers. In light of the reviews (below this email), we would like to invite the resubmission of a significantly-revised version that takes into account the reviewers' comments.

As you can see from the reviews, the reviewers appreciated the clarity of the work in its current form.  However, two of the reviewers have identified ways in which additional connections with biological realism are needed to raise the significance of this work.  The authors should pursue these steps while taking care to maintain the clarity of the manuscript.

We cannot make any decision about publication until we have seen the revised manuscript and your response to the reviewers' comments. Your revised manuscript is also likely to be sent to reviewers for further evaluation.

Sincerely,

Jonathan Rubin

Associate Editor

PLOS Computational Biology

Kim Blackwell

Deputy Editor

PLOS Computational Biology

Reviewer's Responses to Questions

**Comments to the Authors:**

Reviewer #1: This paper uses computational methods to thoroughly answer a simple but very important question. At what Egaba values and under which conditions is GABAaR mediated synaptic signalling excitatory (ie drive AP generation) versus inhibitory (reduce the likelihood of AP generation)? I am very enthusiastic about this paper and think it is an important contribution to our understanding of GABAergic synaptic transmission. I must say I haven’t felt this satisfied after reading a paper for quite a while. It is elegantly simple, and very satisfying in its conclusions. The major conceptual advance is that the authors come up with a metric ‘EgabaThr’, which defines the Egaba value at which GABAaR mediated synaptic transmission mediates excitation. They then use this to show some intuitive, but important results including the fact that EgabaThr is often actually negative of the action potential threshold. This is particularly the case for two conditions which they demonstrate beautifully: 1) when GABAaR synapses come before AMPA inputs in time, or 2) when GABAaR inputs are more distal to inputs AMPA inputs in space. They also demonstrate that for low conductance tonic GABAaR, EgabaThr is more negative than the action potential threshold. These are all important insights and the field is richer for them. The simulations are appropriate and well performed, the manuscript is well written and logically developed I therefore endorse its publication. I only have two minor comments:

1) Although I presume this was the case, can the authors confirm that active mechanisms (ie voltage gated Na+ and K+) only in the somatic compartment and that the dendrite is passive, and make this explicit in the methods. Further when authors explore the effects of varying the spatial position of the GABAaR synapse on the dendrite the authors are determining the effect on excitability as measured by AP generation at the soma. In the discussion it may be worth the authors acknowledging this somatocentric viewpoint of GABAergic inhibition especially as in many types of neurons (including the CA3 pyramidal neurons modelled here) that dendrites host active conductances and that non-linear input summation and integration also occurs within the dendritic tree itself. This was not modelled here and this should be acknowledged as it is likely that an effect of distally targeted dendritic GABA is to control local active conductances in the dendrite itself (a dendrocentric viewpoint), which could complicate the picture a bit. I’m not suggesting the authors model active dendritic conductances but if they could perhaps acknoweledge this point in the discussion that would be helpful

2) I believe in Line 440 “ gGABAThr “ should be “gampaThr”

I congratulate the authors again on an excellent paper.

Reviewer #2: The authors investigate the value of EGABA at which a GABA synapse triggers an action potential in a simple in silico models. The authors investigate the important relations between this threshold value, the GABA conductance, the presence of an AMPA synapse, the tonic GABA conductance and the distance between the GABA synapse and the soma. They found importantly that there are complex relations between the GABA threshold value and all of these parameters.

OVERALL APPRECIATION: The paper is well written and the methodology is sound. The authors share their code which seems to me to be ok. The questions tackled in this work are important and of interest. However, it is difficult for me to appreciate to biological implications of this work due in part to the rather simple nature of the model. For example, the GABA conductances mentioned in the paper are sometimes order of magnitudes larger than realistic values for GABA synapses. As there is a lot of variability from cell to cell and as membrane potential fluctuates in time, it is not clear to what extent it is relevant to give a precise value for the threshold value of EGABA. What would be the results if the authors added an axon in the model and assumed that the GABA synapse is in the axon initial segment? What would be the results in a more realistic scenarios of several randomly distributed synapses?

Specific comments:

1) In figure 1, The action potentials in A and B look rather different with the simulated train of AP exhibiting after spike hyperpolarization which seems to be absent in the recorded trace. How to explain this difference?

2) Line 174. The authors mention 'several hundred sweeps'. Could they explain a bit more? Is it because of the Markov modelling of the spike generating channels or to sweep accross all of the parameter values? What is the impact of the number of sweeps and how did they author determined this value?

3) In fig 2, you have conductance values in several hundreds of nS, this range seems exagerated to me as gaba synaptic conductance should be in the order of 1nS.

4) Did you try any topology beyond the ball and stick one? What do you believe would be the impact of a more complex topology?

5) In fig 3 E and F, I think we see the limit of the numerical precision as the lines seem quite irregular especially in 3 F. Is the 'noise' indeed due to numerical limitations or is there something else at play, please discuss.

6) I find it difficult to distinguish between the different shades of blue and orange in figure 4. I suggest using a color scheme for which it is easier to distinguish between the different colors. The same goes for figure 5 B.

7) I find it difficult to understand why the Hodgkin-Huxley is not suitable here? Could you discuss and justify more your choice for the model of voltage gated Na+ and K+ channels?

8) A conductance ohmic equation is used to compute the current through Na+ and K+ channels. Is that the same for the Cl current? If so, why not use the GHK flux equation which explicitly takes the intracellular and extracellular concentrations of Cl- and HCO3- into account? The results might not be exactly the same and it could be more relevant to relate the investigations to Cl- concentrations instead of EGABA.

I wish the author good luck and acknowledge that a lot of work went into the writing of this manuscript.

Reviewer #3: In neonatal brain, the GABAergic synaptic inputs exert excitatory effect unlike the matured adult brain where GABA causes inhibition. This reversed role of GABA is due to the higher intracellular Cl- concentration caused by the low expression levels of K+-Cl- cotransporters that extrude Cl- from the cell and the accumulation of Cl- due to NKCC1. This leads to the reversal potential for GABA (EGABA) to be less negative than the resting membrane potential. This study explores additional conditions required for GABAergic synaptic inputs to be excitatory. The key conclusions from the study are that for GABAergic inputs close to soma to be excitatory the threshold EGABA above which GABA becomes excitatory is close to the threshold for action potential. This threshold EGABA shifted to positive values. This threshold EGABA also depends on the spatial and temporal relation between GABA and AMPA inputs where it shifted to more negative values for AMPA inputs appearing after GABA input. The threshold EGABA shifted to values negative to threshold for action potential when AMPA synapses located proximally to the GABA input, while for distally located AMPA synapses the dendritic distance had only a minor effect on the threshold EGABA. Thus, the study shows that the excitatory effect of GABA is more complex than just the change in Cl- dynamics in neonatal brain.

Overall, I find this study very interesting and worth considering for publication in PLoS Comp Biology. However, a few issues remain that need to be addressed.  

In the first two postnatal weeks, the reversal potential for GABA is significantly more negative than the range predicted by the model. That is, the observed EGABA < -50 mV (see for example, Owens et al. J Neurosci. 1996, 6414-6423) , whereas the model predicts that the threshold EGABA above which the GABA is excitatory is greater than or equal to -43.1 mV. By this account, the experimentall observed EGABA should always be inhibitory, which is not the case. Thus a detailed discussion of the model-predicted EGABA threshold in light of the experimental observations is needed.

A model closely reproducing the shape of AP (that is, the sharp rise phase of AP) was previously developed by Naundorf et al. (2006) Nature 440(7087):1060–1063. Why is a new (more complex) formalism needed to model the actual shape of AP?

AMPA synapses are not developed in the first few postnatal days (see for example, Lohmann and Kessels, J. Physiol. (2014) 592, 13-31). Which raises the question about the validity of the simulations involving simultaneous AMPA and GABA inputs. The authors need to discuss the postnatal ages at which the different simulations performed in this study are relevant.

Following are some minor concerns.

L 38: “immature brain of after neurological insults” should be “immature brain or after neurological insults”.

L 39: “depolarizations con contribute” should be “depolarizations can contribute”  

L 40: “to determine which amount of a GABAergic” should be “to determine what amount of a GABAergic”.

L 78-79: The statement “However, it is important to consider that depolarizing GABA responses do not per se lead to excitatory effects” is not clear. GABA receptors are known to release Cl- from the cytoplasm in the neonatal brain, leading to excitation. Besides, if GABA responses are “depolarizing” then why won’t they lead to “excitatory effects”.  

L 169: The sentence “the distinct EAPThr parameters are virtually independent on the duration of” should be “the distinct EAPThr parameters are virtually independent of the duration of”.  

L 242: “illustrated that gGABAThr showed a considerable less steep dependency” should be “illustrated that gGABAThr showed a considerably less steep dependency”.

**Have the authors made all data and (if applicable) computational code underlying the findings in their manuscript fully available?**

Reviewer #1: Yes

Reviewer #2: Yes

Reviewer #3: Yes

PLOS authors have the option to publish the peer review history of their article (what does this mean?). If published, this will include your full peer review and any attached files.

Reviewer #1: **Yes: **Joseph Raimondo

Reviewer #2: **Yes: **Nicolas Doyon

Reviewer #3: No
---

## [Decision Letter · Decision Letter 1]

24 Oct 2021

Dear Dr. Kilb,

We are pleased to inform you that your manuscript 'Modelling the spatial and temporal constrains of the GABAergic influence on neuronal excitability' has been provisionally accepted for publication in PLOS Computational Biology.

Best regards,

Jonathan Rubin

Associate Editor

PLOS Computational Biology

Kim Blackwell

Deputy Editor

PLOS Computational Biology

Reviewer's Responses to Questions

**Comments to the Authors:**

Reviewer #1: I am satisfied with the authors' revisions and recommend acceptance of the manuscript.

Reviewer #2: Congratulations to the authors. They did an impressive work in addressing all the points raised by the reviewers. I recommand the paper to be published as such.

I have a few very minor comments:

Lines 260-261, its preferable not to break a line in the middle of an equation.

Line 276, the format of EAPThr seems strange. Not sure if it comes from the pdf transformation

Line 439 'is negative to about' is a strange formulation

Line 574 the end of the sentence seems to be missing.

Line 776 and below maybe use L (capital l) for length since lower case l is identical to 1 in this font.

Line 789 and below, punctuation should be used after equations.

Line 832, there is a problem with the format of gGABA

Line 838 gAMPA is in italic but should not be

Reviewer #3: No more comments

**Have the authors made all data and (if applicable) computational code underlying the findings in their manuscript fully available?**

Reviewer #1: Yes

Reviewer #2: Yes

Reviewer #3: Yes

PLOS authors have the option to publish the peer review history of their article (what does this mean?). If published, this will include your full peer review and any attached files.

Reviewer #1: **Yes: **Joseph Raimondo

Reviewer #2: No

Reviewer #3: **Yes: **Ghanim Ullah

---

## [Editor Report · Acceptance letter]

8 Nov 2021

PCOMPBIOL-D-21-01116R1 

Modelling the spatial and temporal constrains of the GABAergic influence on neuronal excitability

Dear Dr Kilb,

I am pleased to inform you that your manuscript has been formally accepted for publication in PLOS Computational Biology. Your manuscript is now with our production department and you will be notified of the publication date in due course.

With kind regards,

Agnes Pap
